# Units of Military Fortification Complex as Phenomenon Elements of the Czech Borderlands Landscape

**Jiří Kupka** [1] , **Adéla Brázdová** [2,*] and **Jana Vodová** [1]

1 Department of Environmental Engineering, Faculty of Mining and Geology, VSB–Technical University of Ostrava, 17. listopadu 2172/15, 708 00 Ostrava, Czech Republic; jiri.kupka@vsb.cz (J.K.); jana.prymusova@gmail.com (J.V.)
2 Department of Building Materials and Diagnostics of Structures, Faculty of Civil Engineering, VSB–Technical University of Ostrava, 17. listopadu 2172/15, 708 00 Ostrava, Czech Republic
* Correspondence: adela.brazdova@vsb.cz

**Abstract:** This paper is focused on selected units of casemates with enhanced fortification in the military fortification complex of the Czech borderlands landscape as specific forms of brownfields. They represent a functional system that interacts with surrounding nature, landscape character, and human society. Four approaches were chosen to study the function and potential of selected individual abandoned casemates with enhanced fortification, where each of them corresponds to one of the four landscape layers: genius loci, socio-economic sphere, functional relationship (between human and the landscape), and natural conditions. There is a corresponding research method for each of the landscape layers (guided interview with respondents, data analysis on abandoned casemates with enhanced fortifications as brownfields, analysis of their landscape functions, and zoological survey of interior). The main results could show that abandoned casemates with enhanced fortifications can play important roles in all landscape layers: stories and genius loci, abandoned casemates with enhanced fortification as a special type of military brownfield but also as a semi-natural ecosystem, and the same time as a habitat for invertebrates. The analyses and surveys conducted clearly demonstrate that abandoned casemates with enhanced fortification as units of military fortification complex of the Czech borderlands landscape perform several hidden important functions in the landscape for which they cannot be viewed as brownfields. This hidden functional potential is most likely best described by the concept of hidden singularity, which offers itself for integration into basic approaches to brownfields.

**Keywords:** brownfields; military fortification brownfields; casemates with enhanced fortification; historical and fabricated stories; semi-natural ecosystem; hidden curriculum; butterflies and moths (Lepidoptera); land snails (Gastropoda); hidden singularity

## 1. Introduction

Brownfields in general can represent one of the key environmental problems. Although they may not always be associated with ecological burden, they always interact with the human, the landscape, and with the surrounding nature. These may be sites that are abandoned, underused, but may also be historically or architecturally significant. The regeneration of brownfields is one of the basic strategies for improving conditions not only in the urban environment (regeneration strategies vary across Europe, as does the definition of brownfields itself) [1]. It is the interactions between people, brownfields and their associated stories, landscape, and nature which this paper addresses.

After the departure of the Soviet army and following the fall of the Iron Curtain (in 1991), abandoned military buildings became a major issue in the Czech Republic, e.g., complexes of buildings such as barracks, shooting ranges, and other buildings and lands too [2].

The former line of Czechoslovak fortifications consists of Casemates with Enhanced Fortification (CEFs), heavy fortifications, and artillery forts. All of these units have since

lost their significance and have become abandoned—and they can now be marked as specific types of military brownfields [3]. The same line of Czechoslovak fortifications, also consists of Abandoned Casemates with Enhanced Fortifications (A-CEFs), has become an integral part of the cultural landscape, which itself is the result of millennia of interaction between nature, man, and his activities. The cultural landscape has been influenced by the military landscape.

Similar military landscapes from different periods of history can be found in other parts of Europe (e.g., Vallo Alpino), and consequently throughout the world (e.g., Great Wall of Gorgan) [4,5]. Some relics of post-military landscapes are even included in the World Heritage List (e.g., Atlantic Wall, The Great Wall in China) [6]. Military landscape (more precisely post-military landscape) is the result of the interaction of natural and anthropogenic factors (economic, technical, political, and cultural human influence) that are bound to a specific area with a common history [7–13]. Human military activity affects not only the appearance but also the structure and function of the landscape. This is evident in the case of the fortification lines [11,14]. The post-military landscape also becomes part of the evidence of the historical development and therefore part of the cultural heritage of the area [15]. In the case of abandoned military objects (in our study A-CEFs units) the landscape acquires new character and function, possible variability of the use of these objects but an 'atmosphere' connected with these objects 'remains' in them [16,17].

The landscape can be characterized by layers, which we perceive as the result of the relationships of its individual components that change dynamically over time [9,18,19]. We can recognize four layers of landscape—genius loci, socio-economic sphere, functional relationship (between human and the landscape), and natural conditions. **Genius loci** are the spirit of the place, respectively of the landscape [20]. This layer is also the first aspect that causes the interaction of humans with the landscape (evokes emotions). We consider the **'socio-economic sphere'** as a cultural heritage, human creations and their history but also recent use of the landscape, and the spiritual perception of the landscape. The socio-cultural sphere describes the functional relationship of the landscape with humans. The **'Functional relationship'** of the landscape with humans should be defined by the socio-cultural sphere. **'Natural conditions'** include living and non-living nature, including natural processes and occurrences.

Military fortification units represent an immense fortification system of Czechoslovakia (1918–1938). These units were built in 1935–1938, just before the outbreak of the Second World War (WWII), and this line was never fully completed [21,22]. In general, it is composed of a system of strategically placed CEFs units (Figure 1), heavy fortifications, and artillery forts, especially in border areas [22–24]. The whole system of Czechoslovak fortifications was inspired by the model of the Maginot Line, which was a system of fortifications built in France after the experience of the First World War [22,24]. As a result of the Munich Agreement (September 1938), the territory where the Czechoslovak fortification system was located, fell to the then German government [21–23,25]. The fortification system thus failed to fulfill its expected defensive purpose, as after 30 September 1938 all objects were abandoned by the Czechoslovak army and subsequently occupied by German troops [21–23,25]. At the end of WWII, some of the buildings served as strong points for the German army against the advancing Soviet army and were damaged during battle [22,24]. In the post-war period (1945–1989) the objects gradually lost their military significance [24–26]. Only a few segments of the whole Czechoslovak fortification system were renovated for the purpose of building the so-called Iron Curtain, adapted as fallout shelters, or used as storage facilities for military material [25].

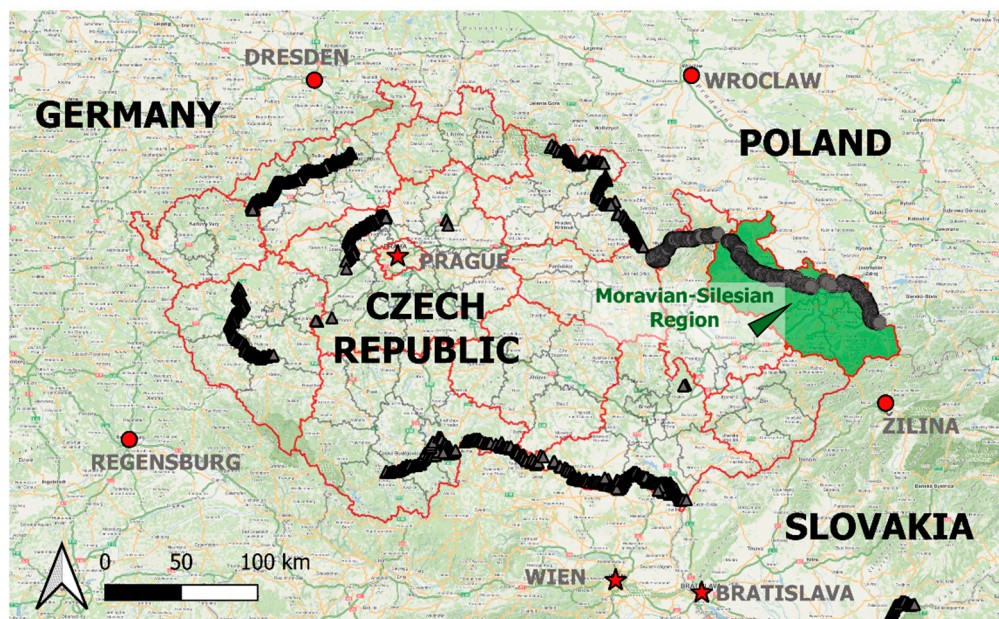

**Figure 1.** CEFs units on the whole territory of today's Czech Republic—noticeable continuing line to Slovakia; triangle marks are CEFs units from 1936, dot marks are CEFs units from 1937–1938 [27–29].

Some of the fortification units were destroyed as a result of devastation during the liberation battles or as an obstacle to technical infrastructure (quarries, transport infrastructure, etc.) [27]. The remaining buildings were used, for example, as warehouses for various materials (e.g., fruits and vegetables, fertilizers, or sprays) or remained abandoned [30–32]. Only a few segments of the military fortification units are protected as a cultural monument and may be marked for our purposes as UA-CEFs (Used Abandoned Casemates with Enhanced Fortification) [33]. Currently, some fortification objects are offered for sale to private ownership [25,34]. We can therefore conclude that as a result of historical events, a significant part of the fortification system built in 1935–1938 became a unique type of military brownfields almost immediately after the end of WWII.

The construction of the Czechoslovakia fortifications complex in the territory of the Moravian-Silesian Region (according to the current territorial division of the Czech Republic) began in the Ostrava region. It was expected that the enemy would make the greatest offensive here. The fortification system was also intended to serve as protection for the industrial area (the so-called 'steel heart of the republic') [24,35]. The complete line of fortifications (CEFs and heavy fortifications) then continued along the border with Germany towards western and southern Bohemia [24,36]. Another line ran through southern Moravia and ended at Bratislava but was originally intended to continue further east to Košice (border with Hungary) [24,36].

The subject of this study, as an example of selected Military fortification units as specific forms of brownfields, is an A-CEF 'model 37' [27,32]. This construction is reinforced concrete with a front wall and ceiling thickness of 80 or 120 cm (normal or reinforced modification) [27,32]. From the direction of the expected enemy attack, the fortress was additionally provided with an embankment made out of boulders and covered with a layer of earth and grass, which further strengthened and camouflaged the object [22,32]. The entrance to the building consisted of a bar and one armored door (at right angles to the bar) [22,27]. Over time, five infantry types were designed in three basic levels of resistance, which could be used to protect any terrain without the need to build atypical solutions. This system simplified, accelerated, and reduced construction costs thanks to the possibility of using standardized internal equipment [21,25,27]. At the same time, the building was equipped with an entrance loophole, grenade chutes, and one or two periscopes in the ceiling of the fortress [25]. The crew was made up of 4 to 6 men, while the size of the interior space was about 8 m$^2$ [32].

These objects as brownfields represented by the Czechoslovak fortification units (A-CEFs) are an integral part of the Czech border post-military landscape. Therefore, we can also speak of the objects of the Czechoslovak fortifications as a phenomenon of the post-military landscape. A part of the post-military landscape with A-CEFs is shown in Figure 2.

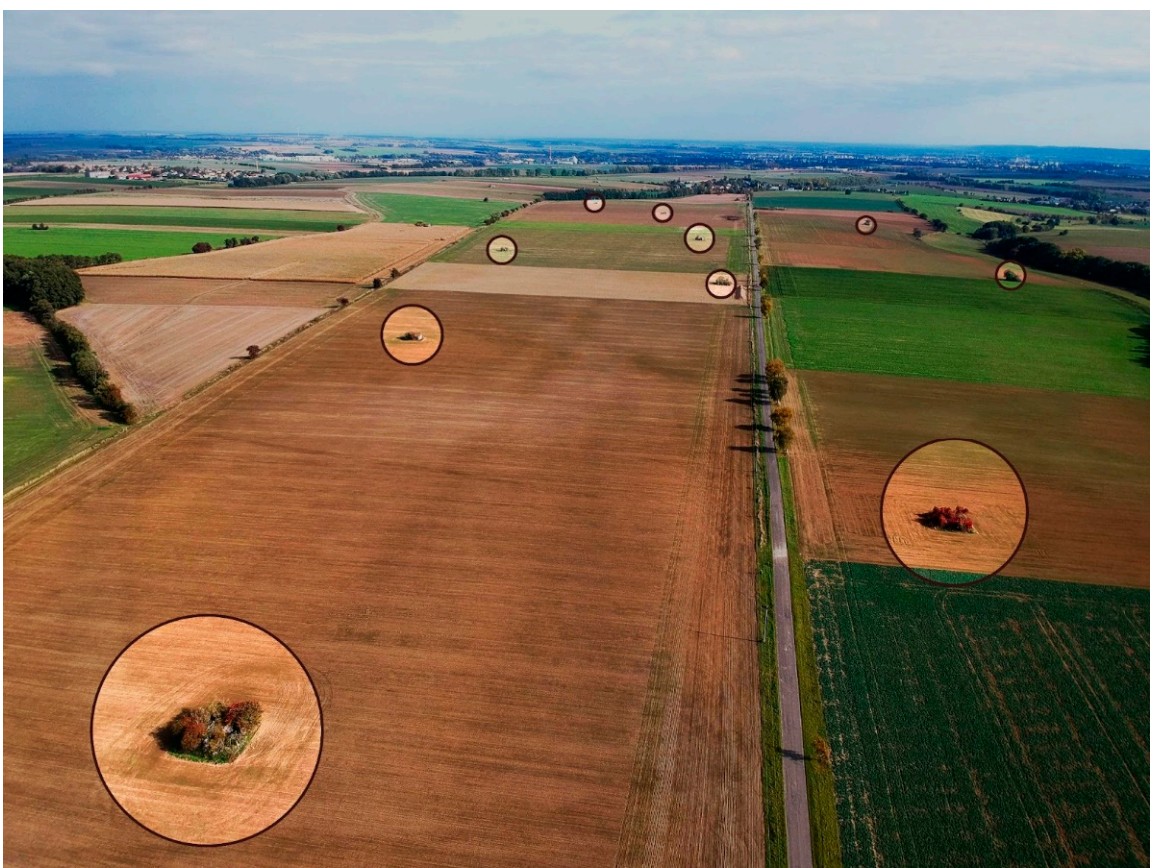

**Figure 2.** Visible line of Czechoslovak fortification complex (A-CEFs) in part of Moravian-Silesian Region, on part of the section 'Milostovice'; photographed from drone.

The aim of the study is to provide a conceptual approach to A-CEFs as specific types of brownfield, while introducing and describing the roles which A-CEFs play in the landscape and how these roles relate to the definition of brownfields. For this reason, four approaches were chosen to study the function and potential of individual A-CEFs. Each of these approaches corresponds to one of the layers that can be recognized in the landscape [37]:

The first approach corresponds to the landscape layer of **genius loci** of the place—the aim of the search is stories [38]. Our aim is to determine whether or not A-CEFs fulfill this function. At first glance, it is evident that A-CEFs have lost their original function. Although they were abandoned, they can still play an important function in the hidden curriculum of the landscape. The term hidden curriculum is borrowed from the field of education. This refers to the hidden lessons of education that are taught by the school and that do not follow the official plans and intentions of the school system or teachers (as opposed to the regular "curriculum", which is the official content of education in the broader sense). The hidden curriculum of the landscape refers to activities of an unofficial character (e.g., off-trail walking, camping outside designated areas, entering bunkers), as opposed to official use (e.g., in the context of tourism, use of conventional accommodation, walking on designated hiking trails, etc.) [39]. In some cases, the hidden curriculum may also be illegal (e.g., entering abandoned mines, abandoned buildings).

The second approach corresponds to the **socio-economic sphere** of the landscape layer (human creations, their history, etc.) [37]. In our case, we select anthropogenic elements

from the post-military landscape, which at first glance have lost their function and therefore meet the definition of a brownfield. On the other hand, the first approach shows that A-CEFs can still perform certain functions in the landscape (hidden curriculum). From the point of view of this approach, these objects have their historical value. Not only UA-CEFs (e.g., as museums), but also A-CEFs represent specific elements in the landscape that can perform certain functions but at the same time fall into the category of military brownfields. The problematic question of the second approach is: Are A-CEFs 'real' brownfields and is their remediation therefore necessary?

The third approach corresponds to the **functional relationship between human creations and landscape** as layers [37]. In our case, we are focusing on a landscaped enclave influenced by human activity. This enclave is directly formed by the A-CEFs or UA-CEFs themselves and their immediate surroundings. To identify the maximum possible use of anthropogenic elements in the post-military landscape is necessary to look for their function and potential. The problematic question is: What is the significance of these objects in landscape interactions?

The fourth approach corresponds to **natural conditions** (living and non-living nature, natural processes, and occurrences) as the landscape layer [40]. In our case, this involves obtaining biological data from a field survey. We look at post-military sites (A-CEFs as brownfields) not only as purely anthropogenic habitats but as semi-anthropogenic to natural habitats. Invertebrates, which are expected to be present in the interior of A-CEFs [41,42], were chosen as a model group of organisms. The problematic questions, in this case, are: Which species occur in bunkers, and what is the nature of their distribution in these objects? Also, which environmental factors may the distribution of these species depend on?

## 2. Materials and Methods

The Czech Republic is located in the heart of Europe, bordering Germany, Poland, Slovakia, and Austria. The Czech Republic is divided into 14 regions. The area of interest for the purposes of this paper is the border region of the Moravian-Silesian Region, where the A-CEFs or UA-CEFs under study are located. The Moravian-Silesian Region is situated in the eastern part of the territory of the Czech Republic. The northern and north-eastern part of the region borders Poland (today's border does not correspond with the pre-WWII state border), and the south-eastern part of the region borders Slovakia. The region also borders on the west and south-west with other territorial units—the Olomouc Region and part of the southern border is adjacent to the Zlín Region. The geographical situation of the Moravian-Silesian Region is shown in Figure A1.

Throughout former Czechoslovakia (1918–1938), almost 12,000 CEFs were built (or planned), as well as almost 1000 units of heavy fortifications (artillery logs and forts), which are not the subject of this study [25]. In the Moravian-Silesian region alone (in terms of the current administrative structure of the Czech Republic) 896 CEFs were planned [27]. The continuity of the whole line in the Moravian-Silesian Region is shown in Figure A1. In this figure, there is also a clear continuity with the line in the Olomouc Region. Due to a large number of these objects, only some of them were selected for the purpose of a more detailed study.

Geology and geomorphology played a significant role in the construction of CEFs [35]. It is worth noting that the Moravian-Silesian Region is covered by two geological units, namely the Bohemian Massif and the outer Western Carpathians [43,44]. The development of these two geological units is complemented by quaternary sediments whose origin is linked to continental glaciation, which left deposits of gravels and sands [43,44]. There are all types of relief from highlands and hills to lowlands in the Moravian-Silesian Region [43,44]. As the zoological survey shows, the amount of precipitation and the number of days with snow are crucial. According to Quitt [45], the lowland areas of the Moravian-Silesian Region fall into a moderately warm climatic area, while the mountain and foothill areas fall into a cold climatic area. In relation to the biogeographical classification of the Czech Republic,

the territory of the Moravian-Silesian Region is part of the Central European deciduous forest province (like the vast majority territory of the Czech Republic), where parts of three subprovinces meet—the Hercynian subprovince, the Polonian subprovince and the West Carpathian subprovince (simplified in the direction from west to east) [46]. The boundary between the subprovinces is not distinct, and in a large part of the area of interest, it can be characterized as a transitional zone of mutual influence. The fauna here is relatively diverse, which is related to the geographical conditions (mixing of West Carpathian, Polonian, and Hercynian elements) [46].

## 2.1. Genius Loci—Layer of the Landscape: Searching for Stories of the A-CEFs

One of the functions of A-CEFs in a post-military landscape is the role they play in specific genius loci [14,47]. In order to capture this potential, a guided interview method was chosen, where respondents were asked about the stories associated with A-CEFs (including capturing the wider context associated with the place). Due to the specific topic of the research, it was necessary to approach suitable respondents who are in some way affected by the genius loci and the associated function of A-CEFs in the landscape. Priority was given to staff from organizations dedicated to leisure activities for children and young people, as well as owners of buildings and last but not least, military history clubs and historians. The initial part of the guided interview dealt with information about the respondent such as age, relationship with WWII history (work/free-time activities/education, etc.), when they first heard about this issue, or if they are visitors/owners of the bunker. As part of the guided interview, respondents were asked two sets of questions with a series of supplementary and extension questions: Q1. Do you know a story associated with A-CEFs (when it happened, where it happened, etc.)? Q2. What do you think about the story (truthfulness, authenticity)? These questions were asked in such a way as to make it clear that the focus of the research is on stories associated with A-CEFs as brownfields and not historical stories (e.g., associated with direct participants in historical events that are associated with A-CEFs serving their original purpose). Respondents were contacted either in person or via electronic communication. Guided interviews were recorded on a dictation machine in the case of in-person interviews. Subsequently, the results of the interviews were transcribed and analyzed.

## 2.2. Socio-Economic Sphere—Layer of the Landscape: A-CEFs as Brownfields

For the purpose of our study, the line of A-CEFs (or UA-CEFs) in the Moravian-Silesian Region which stretches from west to east and follows the current northern state border with the Republic of Poland were selected. The entire line of Czechoslovak fortifications (A-CEFs, heavy fortifications, artillery forts) in the Moravian-Silesian region is represented graphically in Figure A1. In this output, the A-CEFs units were marked according to their structural and technical condition (existing—green, destroyed—orange, initiated—purple, obliterated—blue, unbuilt—grey) and are supplemented by heavy fortifications units (red).

This region was chosen because the authors are familiar with the local terrain and because some of the fortifications were among the most completed before the start of WWII. Furthermore, at the end of WWII in 1945, they played an important role during the war between the German and Soviet armies, and after the war, they were not used (the objects on the border with Poland did not become part of the so-called Iron Curtain). They were abandoned and accessed for casual visitors in the frame of the hidden curriculum of the landscape. Due to the large number of existing A-CEFs on the territory of the Moravian-Silesian Region, the selection was made in such a way as to take into account the greatest possible heterogeneity of the selected objects. This includes altitude (highest and lowest positions of the Moravian-Silesian Region, middle positions), location within the Moravian-Silesian Region (the most eastern and the most western), and also the character of the surrounding environment (open, semi-open, and closed exterior environment).

In order to select a suitable sample of fortification units (A-CEFs) for our study, we used data from available historical military maps first depicting the line of fortifications

throughout former Czechoslovakia. The original assumption was to divide the line in the territory of today's Moravian-Silesian Region into 10 groups/sections (A-J). Within each group, a selection of 5 existing A-CEFs in different environments (forest/forest edge/arc) were considered. While searching for more detailed information for the selection of the 10 groups, websites of friends of military history were found which contained, among other things, databases with more detailed information—for example, information regarding the technical status. These websites were 'The Interactive Map of Czechoslovak Fortifications 1935–1938' and 'Information on Light Fortifications 1936–1938' [27,36]. A search of these databases revealed that one of the originally intended groups A-CEFs had not been built or the fortifications had been obliterated (this group of buildings was, of course, excluded from the further investigation) Figure A2.

During the field survey, some of the buildings were inaccessible (locked or walled entrances, located on private fenced land, flooded with water, and exceptionally, could not be traced in the field). For these reasons, this methodology has been partially abandoned and the distribution of the surveyed Selected Abandoned Casemates with Enhanced Fortification (SA-CEFs) within the Moravian-Silesian Region results in a less than even distribution. It was possible to include the highest A-CEFs as well as the most eastern and the most western A-CEFs in the SA-CEFs. The selection of 39 SA-CEFs is sufficient for further data analysis and can be suitably supplemented or extended in the future. The distribution of individual SA-CEFs in the Moravian-Silesian Region is shown in Figure A3.

Subsequently, each SA-CEF was categorized according to its geomorphological location (WGS-84), climatic conditions, altitude, administrative section according to the former military administration (original military markings, military section assignment, military numbering), and current ownership. The numbering of individual units does not correspond to the order of data collection but was assigned retrospectively. Within each of the 39 SA-CEFs were in field survey semi-quantitatively detected the orientation of the entrance in relation to cardinal directions, the character of the interior environment (dry/wet/flood), the presence of organic and inorganic material in indoor spaces (none/little/lot), human use (unused/occasionally used/intensively used), accessibility of the entrance were also monitored in each unit (open/semi-open/closed) and the type of exterior environment (open/transitional/closed). The expected output is a graphical representation of the corresponding sections. These 39 SA-CEFs will be further processed not only for brownfields issues but also for the zoological survey.

### 2.3. Functional Relationship—Layer of the Landscape: Determination of Functional Potential of A-CEFs

The aim was to present a way of looking at these specific brownfields from the perspective of this approach of the landscape—in terms of the interactions between the objects and the surrounding environment [48]. The question is whether and how these different layers in the post-military landscape interact with each other and what significance they play in the landscape. From this perspective, in the case of the A-CEFs, it is true that they have not been studied at all yet. For this reason, the analysis was not based on empirical data, but on the methodology of the initial approach (especially field observations) to these objects.

For the purposes of our research, we have decided to consider the objects of A-CEFs in this post-military landscape in two specific ways, namely:

(a)　a semi-natural ecosystem of the external environment (consisting of the A-CEF object itself and its immediate surroundings; analogy with rock),

(b)　the semi-natural ecosystem of the A-CEF's internal environment (analogy with a cave).

This approach required a field investigation of individual A-CEFs and the definition of a model of individual objects of the post-military landscape.

We further assume that individual A-CEFs can impact us as natural elements under certain conditions. The expected output of this research is then a graphical representation with the naming of the different parts of these post-military landscape elements observed from different perspectives.

*2.4. Natural Conditions—Layer of the Landscape: Zoological Survey of the Interior Environment of the SA-CEFs*

The aim of the zoological survey was to carry out an inventory of invertebrate fauna in A-CEFs. As this is the first approach of this character to the SA-CEFs, the zoological survey was simplified to the extent that a) only indoor areas were studied, and b) only during the winter period (February/March 2014). This ensured that the number of potential taxa found was minimized as much as possible. Findings were expected of invertebrate species that are able to survive in similar types of environments for long periods of time (e.g., cellars, adits), but which usually live in suitable habitats outside this type of environment (so-called troglophilic or stygophilic species), as well as invertebrates that seek out similar environments for hibernation (hibernation). Probably the most representative-rich group were the expected so-called accidental guests of indoor spaces. Finds of invertebrates very closely adapted to living in underground spaces (troglobionts and stygobionts) were rather not expected.

Collecting was carried out in 39 SA-CEFs. Invertebrates of indoor spaces of the military fortification complex were studied by using conventional flashlights. Recorded species were examined on the walls and ceilings (spiders, butterflies, and moths) or on the floor under various objects like stones, remains of wood (snails, isopods). Collecting techniques as grids or kick sampling methods here were not used. In several cases, it was necessary to collect specimens for further determination by using entomological tweezers or an exhauster. Specimens were fixed in 70% alcohol or killed by vapors of ethyl acetate.

Subsequently, a partial objective of the faunistic survey was to select a suitable model group of invertebrate animals and characterize it with selected diagnostic features of zoocenoses (abundance, dominance, and frequency) [49]. Based on their abundance, we also performed inter-comparison of the SA-CEFs using multicriteria analysis. Selected independent environmental variables were included in the overall analysis. The environmental variables included altitude, humidity conditions inside the SA-CEFs, rate of human use, presence of organic material in the indoor environment, accessibility of the entrance, and the character of the SA-CEF's surrounding exterior environment. The collected data were processed using the R 4.0.5 program, calculating the similarity between the SA-CEFs using the Bray—Curtis index, the distribution of each sampling area depending on the selected environmental parameter using multivariate data analysis (MDS). SA-CEFs, where no live individuals were found, were not included in the analysis.

All data were recorded in a Microsoft Excel spreadsheet. Map outputs were processed in QGIS and zoological data and graphical outputs were processed in R 4.0.5.

## 3. Results

During the implementation of the research aimed at finding the functions that SA-CEFs respectively A-CEFs fulfill in the landscape, a set of results was obtained from all determined four approaches.

*3.1. Genius Loci—Layer of the Landscape: Searching for Stories of the SA-CEFs*

In total, 27 respondents were interviewed. Obtained data do not allow for a more general evaluation, which does not play a significant role in our case. Given that SA-CEFs represent anthropogenic elements in the post-military landscape with a specific history and therefore place with unique genius loci, our aim was to find stories that would capture this genius loci and thus essentially underline the function that A-CEFs fulfill not only in the first approach (Figure A4).

Most respondents had difficulty recognizing between stories that are historical and stories that relate to the site and its function as a brownfields site. Stories whose origins apparently date back to the post-war period were perceived by some respondents (historians, members of military history clubs) as unfounded, apparently fictional. The most frequently recurring motif in the category of unfounded stories was a military-themed plot

set in the WWII period. These stories or even 'fairy tales' were looked upon with disdain by respondents familiar with the history of A-CEFs.

In general, they put themselves in the role of those who want to prevent the spread of these fallacies, and thus they were also reluctant to share them with us and thus to participate further in their spread. These were mainly history experts, but they had encountered similar 'fallacies' at a younger age before they became experts, and these 'fallacies' were often at the origin of their interest in the history of A-CEFs and similar objects.

Only when asked additional questions did these experts comment on the subject of the stories they described as fictional. These included, for example, stories concerning the existence of vast underground spaces, ammunition stores, archives, underground factories, mass graves, etc.

On the contrary, the respondents from among the leaders of clubs working with children in leisure activities had a rather positive attitude towards unfounded stories and 'fairy tales'. This is due to the fact that these stories (fabulations) present objects in a more interesting (adventurous) framework and thus fulfill different functions in troop games, troop rehearsals, or even ceremonies.

The underestimation of the dangers of amateur inspection of the interior of fortress buildings in general (ignorance of the interior construction design, and therefore basically the pitfalls in the form of shafts, wells, and various ventilation openings) was often mentioned. Furthermore, the presence of homeless people who do not hesitate to use various traps to secure their 'property', or the presence of criminal elements.

The analysis of the statements shows some connection between the respondents and the specific SA-CEFs mentioned in the stories (e.g., interactions the respondent had with the object in early childhood or especially in adolescence). Conversely, some of the statements were of a general character, i.e., a story that can be applied to any object (the recurring motif of the underground in the A-CEFs).

The above results clearly show that A-CEFs fulfill 'hidden' functions with this landscape layer which is associated with genius loci. These functions are educational, cultural, social, etc. but at the same time closely connected to the genius loci of the place and thus constitute part of the hidden curriculum of the post-military landscape. This conclusion, therefore, corresponds to our stated objectives about the role of A-CEFs as a phenomenon of the post-military landscape.

### 3.2. Socio-Economic Sphere—Layer of the Landscape: SA-CEFs as Brownfields

The analysis of available data shows that in the Moravian-Silesian Region a line of A-CEFs was created with a total number of 896 objects, of which 591 are still existing, 40 were destroyed (but the remains of the building are still visible), 16 were initiated but not finished, another 182 were obliterated for various reasons and 67 were planned but their construction was never started (unbuilt). The overall state of A-CEFs (including UA-CEFs) in the Moravian-Silesian Region is presented in Figure 3. This line of A-CEFs (or UA-CEFs) was also supplemented in Moravian-Silesian Region by units of heavy fortifications including artillery logs (which was not the aim of the survey). Brownfield in the built-up area of a municipality can represent an economic and social burden (Figure A4B).

The categorization into individual sections is shown in Table 1. The results obtained in the field survey (2014–2021) based on the proposed methodology (chapter) are presented in Tables A1–A5. These 39 SA-CEFs units and their selected characteristics were further studied for zoological survey purposes.

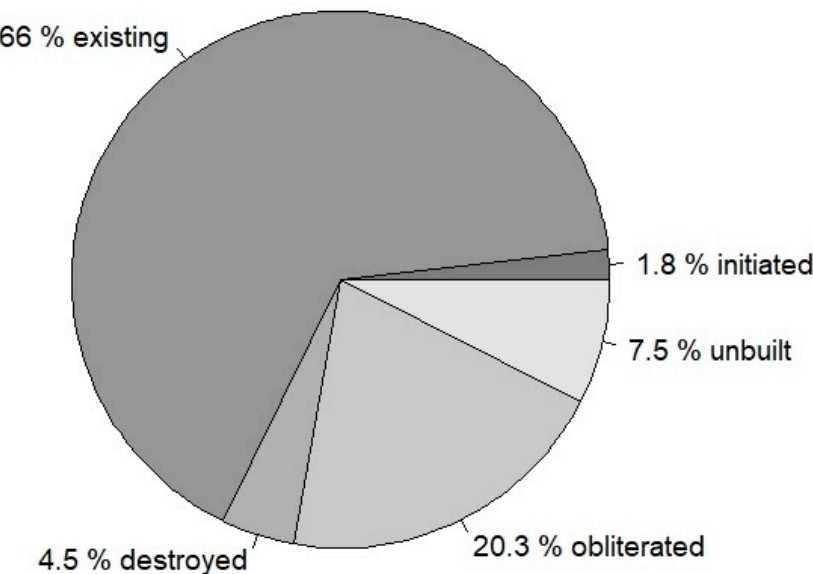

**Figure 3.** Percentage of the current structural and technical condition of A-CEFs (respectively UA-CEFs) in the Moravian-Silesian Region (*n* = 896).

**Table 1.** Summary table of SA-CEFs within each section for further data processing.

| Section: | Total Number of SA-CEFs in Section: | Corresponding Identification of SA-CEFs in the Database |
|---|---|---|
| A | 9 | 2, 3, 4, 7, 8, 9, 37, 38, 39 |
| B | 6 | 10, 11, 13, 14, 15, 16 |
| C | 2 | 5, 6 |
| D | 3 | 33,34 |
| E | 2 | 29, 30, 31 |
| F | 5 | 18, 19, 20, 21, 22 |
| G | 5 | 17, 24, 25, 26, 27 |
| H | 1 | 36 |
| I | 6 | 1, 12, 23, 28, 32, 39 |

Based on our own field survey were found the following rates of human use for each SA-CEF were: 7 units were intensively used, 16 units were occasionally used and 16 units were not used—Figure 4A. The rate of human use was assessed primarily by the presence of artifacts associated with recent and repeated human presence (tables, chairs, lounge chairs, kitchen equipment, food remains, but also activities associated with efforts to restore the property to its original condition, etc.).

The rate of human use of these units is certainly related to the access restrictions. The data shows that most of these units are open and therefore accessible (26), a large proportion is accessible due to overcoming barriers (12), while only one unit of the SA-CEFs is completely closed and therefore inaccessible—Figure 4B. These were, for example, SA-CEFs in front of whose entrance was overgrown with trees, bushes, and/or partially grounded.

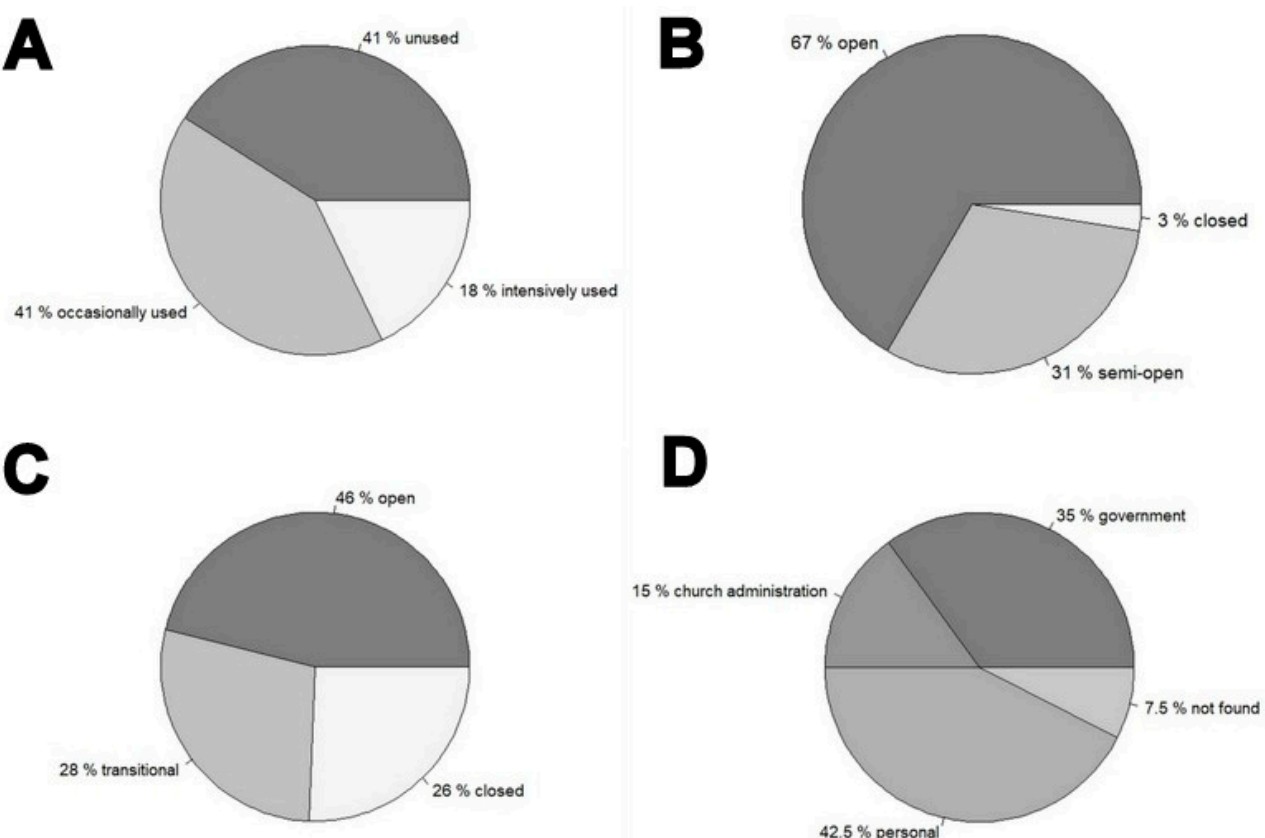

**Figure 4.** (**A**)—Rate of human use of indoor environment in SA-CEFs (*n* = 39); (**B**)—Entrance accessibility in SA-CEFs (*n* = 39); (**C**)—Type of SA-CEF's surrounding exterior environment (*n* = 39); (**D**)—Distribution of SA-CEF's owners (*n* = 39).

Along with the previous two factors—the rate of human use and accessibility of entrance—the location of SA-CEFs units in the landscape may also be related simultaneously. For this reason, a simple analysis of their external environment was carried out and from a total of 39 SA-CEFs, 18 were located in open landscape (meadows, pastures, fields), 10 were located in closed landscapes (forests, scrub) and the remaining 11 SA-CEFs were located in the transitional zone (forest edges, etc.)—Figure 4C.

According to the analysis of property rights, the SA-CEFs were divided into the following categories: government, church administration, personal (physical persons, joint property of married couples, associations, and cooperatives), and the last category is the 'not found' owners—where the SA-CEFs have not been entered in the land register or have an unidentified owner.

From the selected 39 SA-CEFs, only 14 units were owned by the government, and 17 were privately owned ('personal'). In addition, 6 units were owned by a church administration, and 3 units were classified as 'not found' (owner is unknown or owner was insufficiently identified)—Figure 4D. For one of the SA-CEF (number 17), it was determined that part of this SA-CEF is owned by a person and part is owned by the government—for this reason, the data set of Figure 4D is divided into 40 units.

*3.3. Functional Relationship—Layer of the Landscape: Determination of Functional Potential of a-CEFs*

On the basis of the analyses carried out, it can be concluded that individual SA-CEFs (respectively A-CEFs) as anthropogenic elements in the post-military landscape may represent semi-natural elements from a certain point of view. The SA-CEF (or A-CEFs) object itself may be perceived in the landscape at first sight as a 'rock', with a corresponding growth of mosses or vegetation (example shown in Figures 5 and A4B). In contrast, from the point of view of the interior environment, the A-CEFs can be seen as a 'cave', which

can provide shelter for various species of organisms. This 'cave-like' environment of the A-CEFs' interior is not only characterized by a constant temperature throughout the year, but also by calcite deposits and soda straws (as they are called) on the walls and ceilings of the building as 'stalactites'. In terms of geomorphological shapes, the A-CEFs can be perceived as a concave shape that forms an unmistakable step in the terrain. From the point of view of the pedological characteristics, we can consider the object itself as an anthroposol (ceiling, embankment made out of boulders and covered with a layer of earth and grass). In Figure 5 we can observe different perspectives of view as we see the different functions of A-CEFs in the landscape.

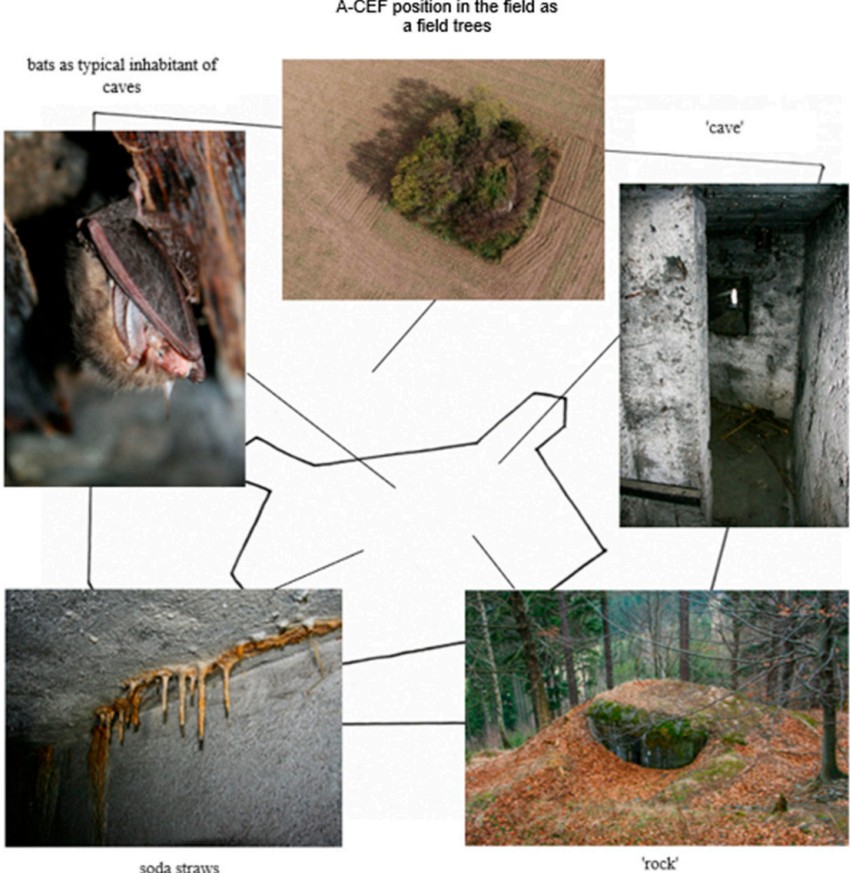

**Figure 5.** Different points of view of individual functions of A-CEFs.

*3.4. Natural Conditions—Layer of the Landscape: Zoological Survey of the Interior Environment of the SA-CEFs*

Representatives of the following taxa were found in the interiors of 39 SA-CEFs: Oligochaeta, Gastropoda (Pulmonata), Arachnida (Opiliones, Araneae, Acari, Pseudoscorpiones), Malacostraca (Isopoda), Myriapoda (Chilopoda, Diplopoda), Hexapoda (Collembola, Diplura, Orthoptera, Dermaptera, Hemiptera, Neuroptera, Coleoptera, Lepidoptera, Diptera, Hymenoptera). The presentation of these results would exceed the scope of this paper. A total of 104 species were identified. Among the taxa mentioned above, Lepidoptera (butterflies and moths), whose adults seek out similar spaces for overwintering or hibernating, and Gastropoda (land snails), which can survive in similar spaces for a long time, or are so-called accidental guests, were chosen as model groups. These two taxa have the advantage of relatively easy and unambiguous determination in the field, with the consequent possibility of determining presence/absence in the SA-CEF and estimating absolute abundance.

A total of 9 species from the model group Lepidoptera were recorded in the interior of the SA-CEFs and a total of 732 live individuals were identified (Figure A5A, Table A6).

From the model group Gastropoda, 20 species were recorded and a total of 180 live individuals were identified (Table A7). The species *Inachis io* represents the butterfly with the highest frequency (91.89%) and the highest dominance (42.76%). The land snail with the highest frequency was *Monachoides incarnatus* (63.64%) and the land snail with the highest dominance was *Helix pomatia* (24.44%) (Figure A5B).

The multicriteria analysis (MDS), which was calculated separately for butterflies and moths, and land snails, does not allow us to interpret the main directions of variability in the species data. Figure 6 shows an ordination diagram (MDS) depicting the distribution of individual SA-CEFs depending on the presence of organic material (land snails) and Figure 7 shows an ordination diagram (MDS) depicting the distribution of individual SA-CEFs depending on the nature of the surrounding environment (butterflies and moths).

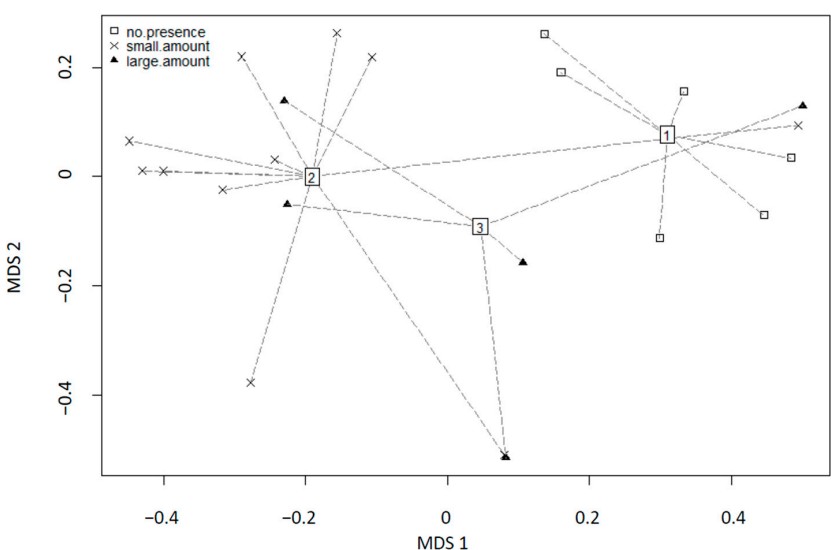

**Figure 6.** Ordination diagram (MDS) showing the distribution of individual SA-CEFs depending on the presence of organic material (land snails, for all SA-CEFs, without data adjustment).

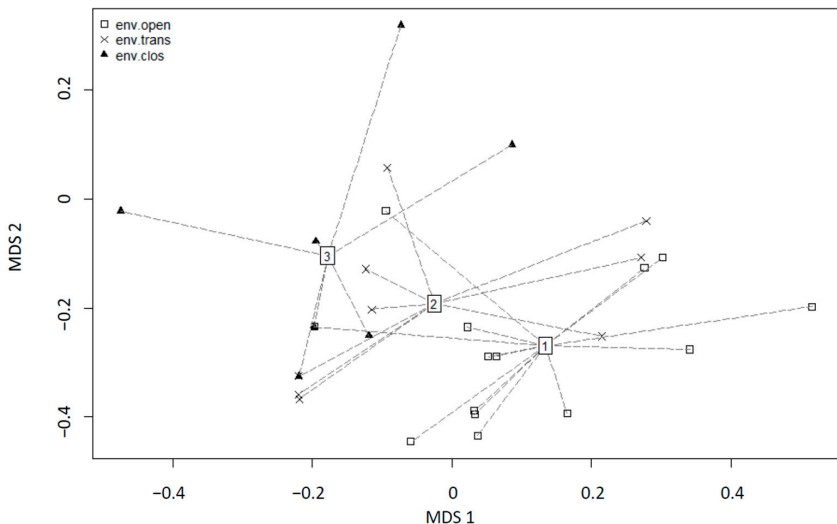

**Figure 7.** Ordination diagram (MDS) showing the distribution of individual SA-CEFs as a function of the nature of the surrounding environment (butterflies and moths, for all SA-CEFs, without data adjustment).

## 4. Discussion

From the partial results that correspond to the individual approaches (in Material and Methods), conclusions can be drawn that will have an impact on further progress in the study of A-CEFs as phenomenon elements of the Czech borderlands landscape.

### 4.1. Genius Loci—Layer of the Landscape: Searching for Stories of the SA-CEFs

The guided interview as a research method was not focused on the general public at this stage, but mainly at people who were expected to have some knowledge of the subject—people who deal with this in their profession (museum workers) as well as hobbyists who are involved in this field in their free time. For the guided interview, which is time-consuming, the number of respondents (27) was not very high, but for our research it is sufficient.

In terms of such a targeted guided interview focused on finding out the stories, neither the ratio of professions nor the age of the respondents is decisive in the results. It is clear from the results that the stories associated with the atmosphere of a place are absorbing regardless of age or profession (but again in a sample of people who have an established relationship with the objects). Both the verbal and written responses from respondents showed their passion for A-CEFs (respectively SA-CEFs) and their enthusiasm for our interest. This is essentially consistent with the function A-CEFs fulfill in the genius loci as a layer of landscape: A-CEFs evoke a deep emotional response in humans. In this context, it is worth quoting part of the statement of one respondent: 'The fortification has absolutely incredible genius loci if you are lucky enough to be hit by it. There are people who are not interested in fortifications, or who are uncomfortable with fortifications for some reason. Then there are the people who are interested in it, who like to read or listen to it. Many visitors leave the sites surprised at all the new things they have learned. And then there are people who, once they've been introduced to bunkers, have never been able to get turned away from them. That's something that can't be described as anything other than their diagnosis.'

These conclusions suggest the usefulness of the method (and results) of the guided interview, but at the same time open the perspective for the creation of a structured questionnaire. A structured questionnaire would be aimed at a broader public and would also take into account the attitudes of people who may not have a positive attitude towards A-CEFs, including the subsequent statistical evaluation of the data. From the results of the questionnaire survey and in the context of the above, it is necessary to appropriately define the conceptual apparatus related to the use of the term 'story', which can be perceived differently within the genius loci as the layer of the landscape (subjective experience of the visitors, etc.) and the cultural heritage (historically documented event, etc.). As already noted in the results, also fabricated stories play an important role in the search for the function of A-CEFs. Given these facts, it is necessary to recognize between different types of stories: historically based stories (historically recorded, more likely to be the accounts of direct participants), historically unfounded stories (not historical fact, but the use of this fact to locate characters, time and space), and fabricated stories (Table 2).

**Table 2.** The role that different story types play in genius loci and socio-economic sphere as landscape layers.

|  | Socio-Economic Sphere | Genius Loci |
|---|---|---|
| **historically based stories** | they play a key role | serve as inspiration to explore on your own; they illustrate the spirit of the place |
| **historically unfounded stories** | can play an important role | can serve as inspiration for various activities |
| **fabricated stories** | their influence is perceived as contradictory to negative | they play a key role |

Although many A-CEFs appear at first sight to be abandoned (A-CEFs as brownfields), they play a role in something that is harder to grasp, and what we might call the 'hidden curriculum of the landscape'. From this perspective, they are not 'really abandoned' but

only as abandoned perform social and educational functions. At the same time, fabulations (apparently fictional stories) play an important role, which is considered worthless, confusing, or even undesirable from the point of view of the cultural heritage. The fact that the casual visitors make up their own stories when interacting with A-CEFs is remarkable when looking at A-CEFs as brownfields, even though a negative phenomenon such as the Goliath effect may be associated with it (Figure A4) [50]. At this point, it is also worth highlighting that, apparently due to the occurrence of SA-CEFs in the open landscape (outside human settlements), we have not observed negative uses (e.g., squatter settlements). But this statement cannot be applied to all A-CEFs, as their very small size makes intensive use rather unlikely.

The SA-CEF units in the landscape were perceived positively by the respondents as a kind of historical milestone reminding them of an important historical stage and may also be part of 'family heritage'. Recording of negative perceptions of the objects is rather to be expected only from the results of a structured questionnaire directed to the general public. This would simultaneously capture the wider variation in the public's perception of post-military landscapes, and hence A-CEFs, for the purposes of our research, and thus also provide a valuable stimulus to a comprehensive view of the issue of A-CEFs as brownfields.

*4.2. Socio-Economic Sphere—Layer of the Landscape: SA-CEFs as Brownfields*

Data analysis has shown that a total of 591 units of A-CEFs and UA-CEFs are still existing in Moravian-Silesian Region (out of a total of 896 originally planned or realized). In 40 cases of A-CEFs and UA-CEFs, it is possible to trace their remains in the landscape. A total of 182 have been permanently obliterated for various reasons. It is interesting to note that a very significant number of all of them have survived and it is questionable whether this is related to difficulties in their obliteration. In certain circumstances, it may be related to their strategic importance within the Czech army for a certain period, or also to the subsequent efforts of leading figures in society (political representation) to preserve these objects as an integral part of the landscape that the Czechoslovak fortification line forms from a historical point of view. Although the study focused only on a limited set of SA-CEFs ($n = 39$), relative to the total number, it is still possible to draw some conclusions of a more general nature.

The ownership of individual A-CEFs or UA-CEFs is complicated in the former Czech Republic due to the complex evolution of property rights. In the set of SA-CEFs, we were interested in the relationship between the type of ownership and the rate of human use. Privately owned buildings were expected to have a higher rate of use, e.g., for recreational purposes or as storage facilities, or to be eventually made inaccessible to the public. The vast majority of privately owned SA-CEFs are not used—Figure 8A.

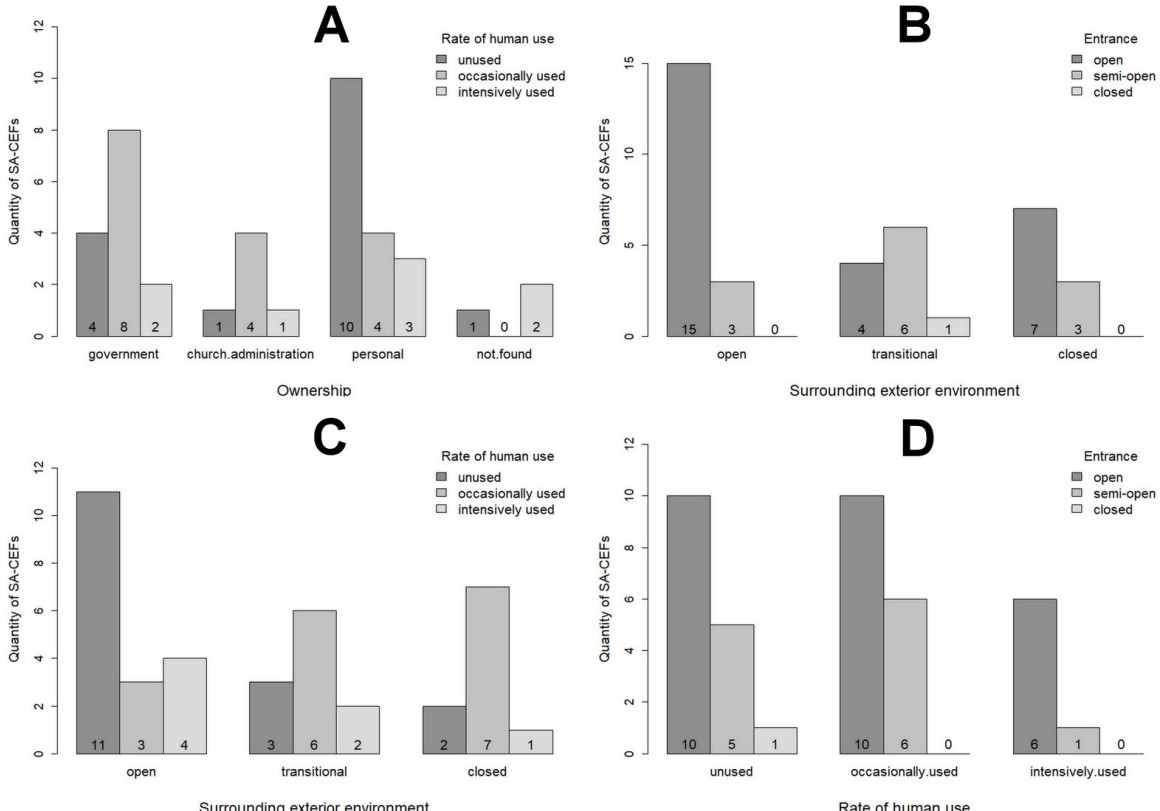

**Figure 8.** (**A**)—Relation between 'Ownership of SA-CEFs' and 'Rate of human use of indoor environment in SA-CEFs'. (**B**)—Relation between 'Type of surrounding exterior environment of SA-CEFs' and 'Entrance accessibility in SA-CEFs'. (**C**)—Relation between 'Type of surrounding exterior environment of SA-CEFs' and 'Rate of human use of indoor environment in SA-CEFs'. (**D**)—Relation between 'Rate of human use of indoor environment in SA-CEFs' and 'Entrance accessibility in SA-CEFs'.

When comparing the type of exterior environment and accessibility of entrance to SA-CEFs (Figure 8B), open landscape SA-CEFs are also generally more accessible. A cross-comparison of the level of human use and type of exterior environment in the SA-CEFs shows (Figure 8C) that open landscape SA-CEFs are the most visited by people. In both cases, this may be related to the possible cultivation of land in the immediate neighborhood of SA-CEFs (one reason for the earlier obliteration of SA-CEFs) versus forest management. The fact that SA-CEFs in open spaces are also more easily found by potential visitors may also be a factor to some extent. Also, when human use rates and accessibility of SA-CEFs are compared to each other (Figure 8D), it appears that logically those SA-CEFs that are also open access are used with the greatest intensity. This comparison also shows that accessibility of the entrance is not a decisive factor for the rate of human use in the sample of SA-CEFs.

*4.3. Functional Relationship—Layer of the Landscape: Determination of Functional Potential of SA-CEFs*

The analyses and results show that there are interactions between SA-CEFs (respectively A-CEFs) and the surrounding landscape, and SA-CEFs (respectively A-CEFs) can be viewed as semi-natural ecosystems. Viewing A-CEFs as a 'rock' or 'cave' (Figure 5 or Figure A4C) offers scope for further research to support this functional potential of SA-CEFs (respectively A-CEFs) in the landscape and further emphasize their importance. At this point, it is at least possible to consider the interior spaces of SA-CEFs (respectively A-CEFs) as overwintering or hibernating sites for some species of butterflies and moths (Results in Section 3.4, Table A6), or also as habitats for various species of invertebrates with different relations to underground spaces. The analysis carried out offers further

possible directions for research (SA-CEFs, respectively A-CEFS, as anthrosols/leptosols, the embankment as anthrosols/cambisols, the walls of SA-CEFs, respectively A-CEFS, as a substrate for calciphilic species of organisms in the acidic environment of mountain forests, etc.). Finally, it is also possible to study A-CEFs from the perspective of island biogeography, where A-CEFs objects can be viewed as terrestrial islands [51]. In this case, it would be appropriate to state that A-CEFs perform ecosystem services in the landscape.

This leads us to the idea that A-CEFs as anthropogenic elements of the post-military landscape has a function (in themselves) and potential in the landscape. However, this potential may not be obvious given that these objects are viewed as classic brownfields (given the definition of brownfields). For this reason, it is appropriate to further address the question of the additional functional potential of A-CEF, which relates to its re-use, but also that it is a hidden functional potential that should be taken into account.

The historical and architectural significance of the relics of the post-military landscape (the fortress lines and their individual objects) are accepted from a global perspective. Therefore, the aim was to point out the functional significance of these objects. This functional significance of these objects can be fulfilled in the landscape itself, regardless of the hidden curriculum of the landscape (based on interaction with humans). The proposed scheme (Figure 5) shows a way of looking at these objects, or rather at their hidden potential. However, the question remains about how to incorporate this view into the definition of brownfields. It can be stated that in our case this approach does not only apply to every A-CEF (in the case of UA-CEFs, its further function is determined by its new use, re-use) but also to other brownfields that may have hidden functional potential. In the case of their re-use may be the hidden functional potential lost. The most illustrative example is the restoration of A-CEF to its wartime condition—cleaning, cleaning the surface, closing all entrances, adding paint, etc. At the same time, it is clear that given their large numbers, not every A-CEF can be restored to this dignified state (as a reminder of the willingness to defend their young homeland for some citizens). Since they are rather small in size (internal area of about 8 m$^2$), they can rather be used only as foundations for huts or for storing domestic crops.

*4.4. Natural Conditions—Layer of the Landscape: Zoological Survey of the Interior Environment of the SA-CEFs*

The largest number of hibernating butterfly species was recorded in SA-CEFs number 14 and 32, a total of 5 species out of 9 species. The largest number of hibernating butterfly individuals was also recorded in SA-CEF number 14 (62 live individuals in total). Only two SA-CEFs had no records (SA-CEFs number 13 and 22). Only 1 and 2 species were recorded in SA-CEFs number 3 and 13. SA-CEFs with such a small number of species recorded also had to be excluded from further analysis (18 SA-CEFs in total). Due to this fact and the very low total number of species found, multivariate data analysis became meaningless.

The most frequently recorded butterflies and moths (Lepidoptera) were *Inachis io* and *Scoliopteryx libatrix*. *Inachis io* from the family Nymphalidae is one of the most common butterflies of opened landscape, but forest edges too. Normally, it hibernates in heatless areas of the building (in attics, in cellars, etc.), and in some cases, it is in the hibernating places even more numerous than the other species, *Scoliopteryx libatrix* [42]. Very common is moth *Scoliopteryx libatrix* from the family Noctuidae, which prefers moist habitats (shores of streams, rivers, and lakes), it occurs also along roadsides, edges of woods, and penetrates well as into the gardens and parks. In autumn, it seeks to hibernate a variety of cavities, caves, tunnels, and cellars [52,53]. It is the most common species of all types of underground space [42].

The largest number of land snail species (Gastropoda) was recorded in SA-CEF number 37, with a total of 9 species out of a total of 20 species. The largest number of land snail individuals was also recorded in SA-CEF number 37 (42 living individuals in total). In total, no records were recorded in 17 SA-CEFs. In SA-CEFs numbers 3 and 10 were recorded only 1 and 2 species. SA-CEFs with such a small number of species recorded also had to be excluded from further analysis. Thus, for the subsequent multivariate data

analysis, it was possible to use data sets relating to only 9 SA-CEFs, and thus, as in the previous case, the multivariate data analysis itself became meaningless.

The most frequently recorded species of land snail were *Monachoides incarnatus* and *Helix pomatia*. *Monachoides incarnatus* from the family Hygromiidae occurs in a variety of forest habitats. It is most commonly found in deciduous forests, in the scrub, often around streams and wet rocks. It also inhabits various secondary habitats such as quarries, gardens, or parks. *Helix pomatia* from the family Helicidae inhabits various types of habitats (especially on limestone), light deciduous forests, scrub, meadows, vegetation along streams, but also cultural areas (gardens, thickets, etc.). In both cases, these are species that can also occasionally be found in different types of underground spaces [41,54].

In addition to 9 species of butterflies and moths, and 20 species of land snails, 75 other invertebrate species from various taxonomic categories have been documented in SA-CEFs. The presence of vertebrates (e.g., bats, rodents) or their resident signs (e.g., marten droppings, bird nests) have also been recorded in SA-CEFs.

The faunistic survey carried out confirmed that the interior spaces of the A-CEFs should be used by various species of animals. Among the species found, for example, spiders (*Meta menardi* or *Metelina merianae*) are able to survive in SA-CEFs for long periods of time, as they inhabit similar habitat types [55]. Butterflies and moths use them for hibernating, and land snails represent so-called accidental guests whose occurrence in indoor spaces is related to the composition of the malacofauna in the immediate surroundings of SA-CEFs. However, some of the land snail species found may also survive in similar types of environments over the long term. These include species such as *Oxychilus cellarius* or *Limax cinereoniger* [41,56]. The occurrence of animals very closely adapted to living in underground spaces (troglobionts and stygobionts) has not been proven. The occurrence of euryvalent species of butterflies and moths (Lepidoptera) and land snails (Gastropoda) is random or in some cases quite rare in the interior of SA-CEFs.

## 5. Conclusions

This paper focuses on A-CEFs in the military fortification complex of the Czech borderlands landscape as specific forms of brownfields. The issue of brownfields as an environmental problem can be viewed from different angles. This corresponds to the different definitions, which are also numerous within Europe [1]. Internationally, the accepted definition is CABERNET, which is based on the CLARINET definition. According to this definition, brownfield sites are areas that have been affected by the previous use of the building, site, and surrounding land. Brownfields are abandoned and under-used, may have real or perceived contamination problems, are predominately located in built-up areas, and require intervention that would enable its continued use.

Based on this definition, the A-CEFs, as part of the military fortification complex of the Czechoslovak borderlands built before the Second World War, are clearly classified as military brownfields. However, these units are very specific compared to other military facilities (in this classification of brownfields). They were not built as an integral part of military complexes or barracks. They were built as separate defensive units, but they all form the whole of a military fortress complex copying the Czechoslovak borderlands. Most of them were built in the open landscape and nowadays form is a phenomenon of the post-military landscape. Referring to the previous definition, are A-CEFs really useless and require intervention to bring them into beneficial use?

For our research, A-CEFs may be technically brownfields, but they are also an integral part of the cultural landscape (post-military landscape) and cultural heritage. As part of cultural heritage, they can enhance social, cultural, environmental, and economic sustainability, preserving diversity and place identity. However, in processing the field survey data, it was found that A-CEFs (respectively SA-CEFs) have further, hidden, functional potential. Our research has shown that A-CEFs as brownfields interact in a specific way with humans (hidden curriculum of the landscape) and, also interact with nature, where they create suitable conditions, e.g., overwintering or hibernating invertebrates, can be considered as

terrestrial islands in the landscape, 'rock', 'cave', habitat and thus can also fulfill ecosystem services. For this reason, A-CEFs cannot, therefore, be seen only as brownfields that need to be f. e. remediated or re-used. Even without such interventions, they have a function and significance in the landscape.

What is the best term to describe the hidden functional potential of A-CEFs? To find a suitable term, it is possible to start from the concept of landscape singularity. This term is used mostly in art and architecture [57,58]. Exceptionally, this term can be used to describe unique features in the landscape as landscape singularities [59]: 'Landscape singularity represents linear or point singularities in the landscape that are natural (watercourse, rock), cultural (urban line, building object) or also historical, but more often a combination of these.' Based on the above characterization of landscape singularity, it can be concluded that the term "singularity" appears to be, in part, the most appropriate name for the character of individual A-CEFs. The common and typical feature of these A-CEFs is that they are all part of the cultural heritage (although they are not listed as heritage sites, e.g., by UNESCO), but for obvious reasons, there is not enough interest in their use (a large number of them). On the other hand, in contrast to the singularity, these sites are "disturbing" in terms of the impossibility of using the site for other purposes (e.g., as arable land), they represent a specific type of brownfield. Their functional potential is hidden and can be discovered only by more detailed study. Although A-CEFs is not currently listed and may be obliterated, under certain conditions they could be left in the context of nature and landscape conservation interests. After all, the analyses and surveys conducted clearly demonstrate that SA-CEFs perform several important hidden functions in the landscape for which they cannot be seen as brownfields. The term that would best describe the hidden functional potential of A-CEFs is a hidden singularity.

The concept of the hidden singularity can be incorporated into the definition of brownfields, where we need to look at them not just as environmental problems, but as environmental opportunities (from a biocentric perspective).

From the outset, this article has challenged the definition of brownfields (in the context of A-CEFs), which is primarily based on the under-use of brownfields or their disruptive interference with the landscape. Considering the results of the research conducted on SA-CEFs, it can be concluded that the concept of hidden singularity can take into account the hidden functional potential and in this case can be included in the classification of brownfields. The hidden singularity of brownfields means that brownfields perform functions in different layers of the landscape (including the hidden curriculum), although these layers may be hidden. Nevertheless, this hidden functional potential can be identified and quantified. From an environmental point of view, the hidden singularity makes brownfields sites that do not need to be revitalized or find new uses for them. Brownfields as an opportunity for investors always assume revitalization or re-use—based on the existing definition of brownfields. In our case, we are discussing the revitalization of A-CEFs to the war form (state of 1937–1938) with a museum exhibition (we can talk about regularity). In the case of re-use, we are discussing, for example, the obliteration of the A-CEFs in order to use the land (irregularity). In the category of brownfields as an opportunity for an investor, the hidden functional potential of A-CEFs could be completely eliminated. The above classification can be applied not only to A-CEFs, but also to other types of brownfields with a similar character. We would like to address the issue of hidden singularity and the hidden curriculum of brownfields in the future.

**Author Contributions:** Conceptualization, J.K., A.B. and J.V.; methodology, J.K., A.B. and J.V.; software, J.K. and A.B.; formal analysis, J.K., A.B. and J.V.; investigation, J.K., A.B. and J.V.; resources, J.K., A.B. and J.V.; writing—original draft preparation, J.K., A.B. and J.V.; writing—review and editing, J.K., A.B. and J.V.; supervision, J.K.; funding acquisition, J.K. and A.B. All authors have read and agreed to the published version of the manuscript.

**Funding:** This research was funded by SGS grant number SP2021/113 and SP2022/122 and the APC was funded by VSB—Technical University of Ostrava—Department of Environmental Engineering,

(Faculty of Mining and Geology) and Department of Building Materials and Diagnostics of Structures (Faculty of Civil engineering).

**Acknowledgments:** We thank our family members for their patience and participation in the exploration of the CEFs.

**Conflicts of Interest:** The authors declare no conflict of interest.

## Appendix A

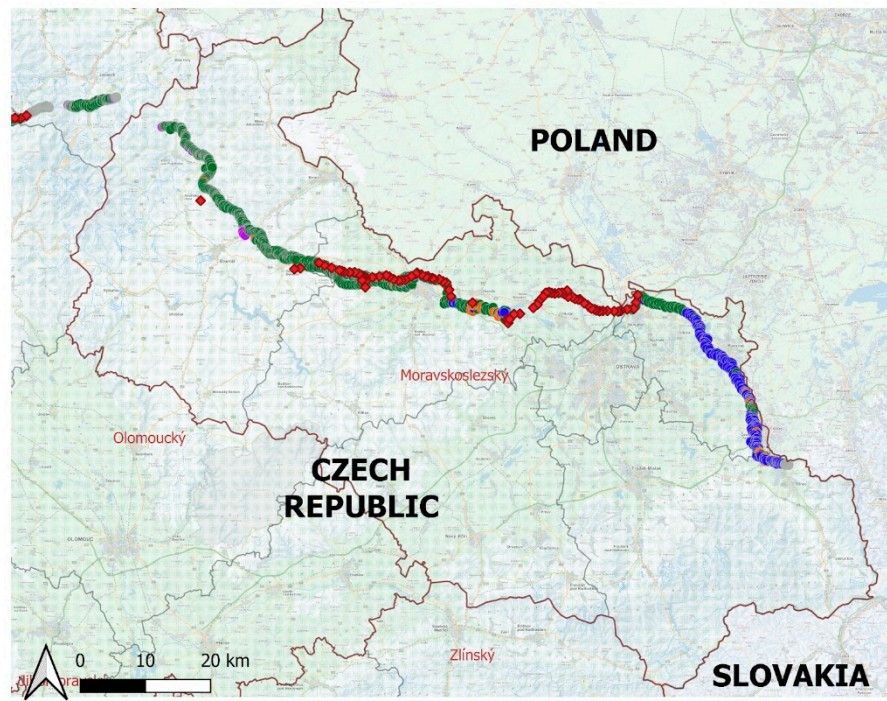

**Figure A1.** Display of the line of Czechoslovak fortifications in the Moravian-Silesian Region—demonstration of graphic interpretation of the obtained data, presentation of the current structural and technical condition of the CEFs using the QGIS program.

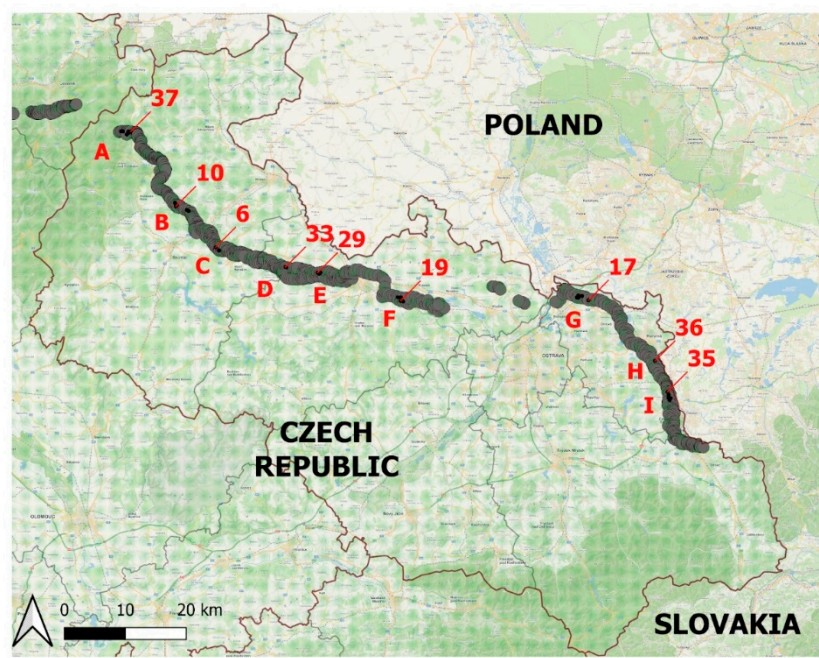

**Figure A2.** Situation of A-CEFs and corresponding sections of SA-CEFs (from A to I) in the borderlands of the Moravian-Silesian Region [28,33–35].

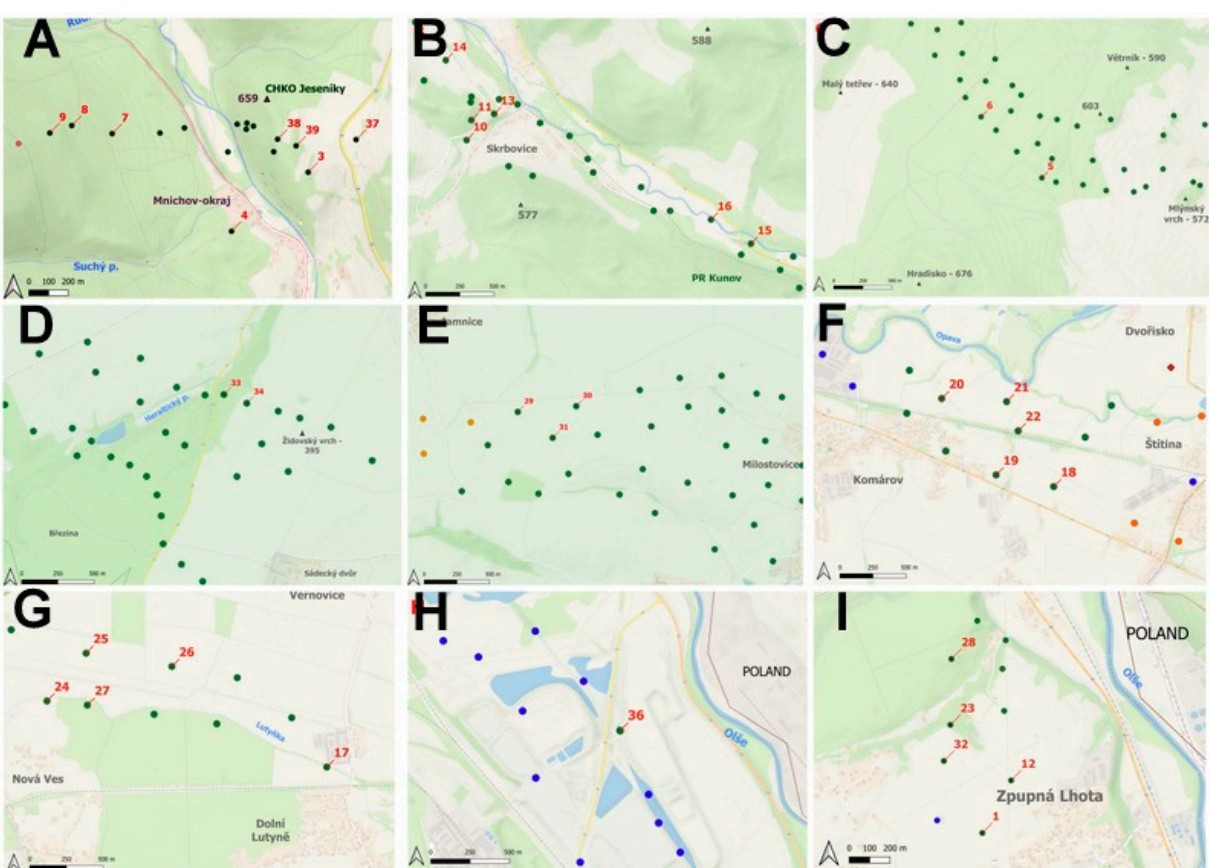

**Figure A3.** Selected parts of fortification line in Moravian-Silesian Region. Individual parts (**A–I**) are shown in Figure A2. Key: green—Existing A-CEF or UA-CEF; pink—Construction was initiated A-CEF; orange—destroyed A-CEF; blue—obliterated A-CEF; red number—SA-CEF.

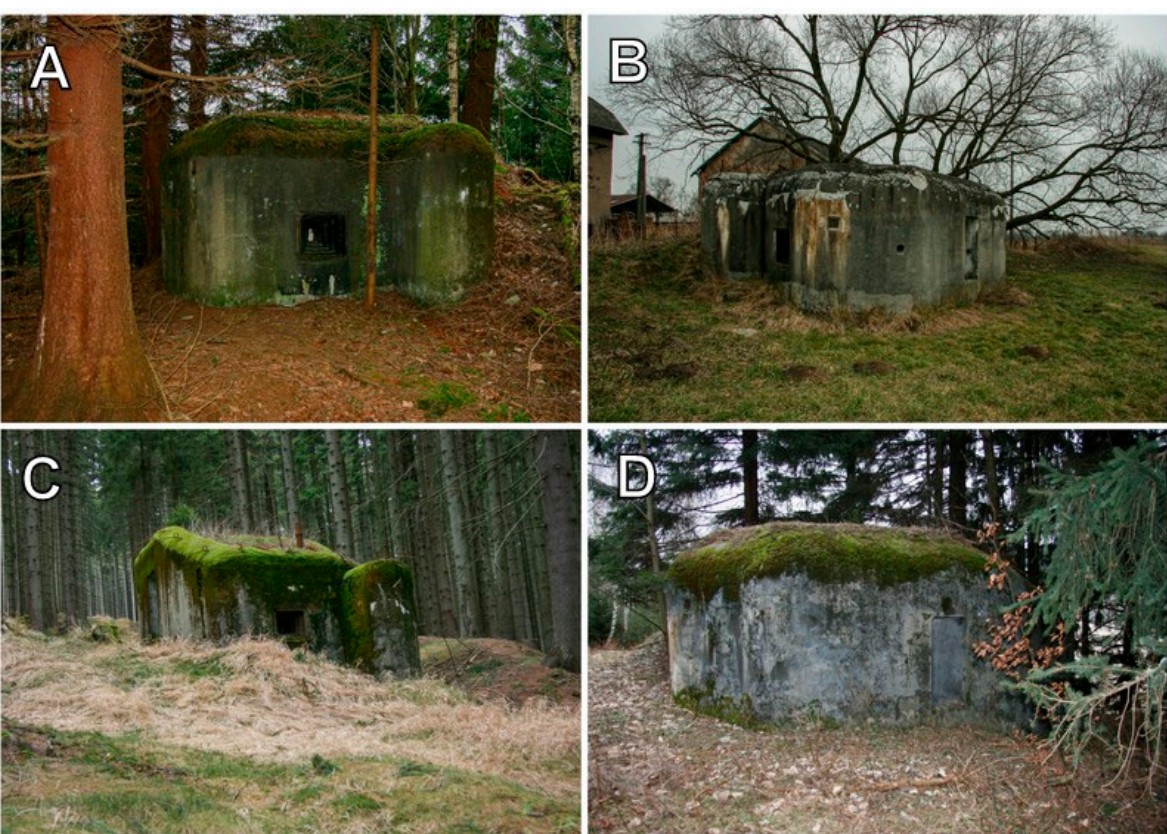

**Figure A4.** (**A**): Each A-CEF object has unique genius loci (e.g., SA-CEF number 38). (**B**): A-CEFs represents a specific type of military brownfields (e.g., SA-CEF number 17). (**C**): SA-CEF number 38 is one of the existing and highest situated objects (715 m above sea level). It serves as an occasional refuge for adventurers and 'could certainly tell' many stories. (**D**): Some A-CEFs can be easily mistaken for rock (e.g., SA-CEF number 39).

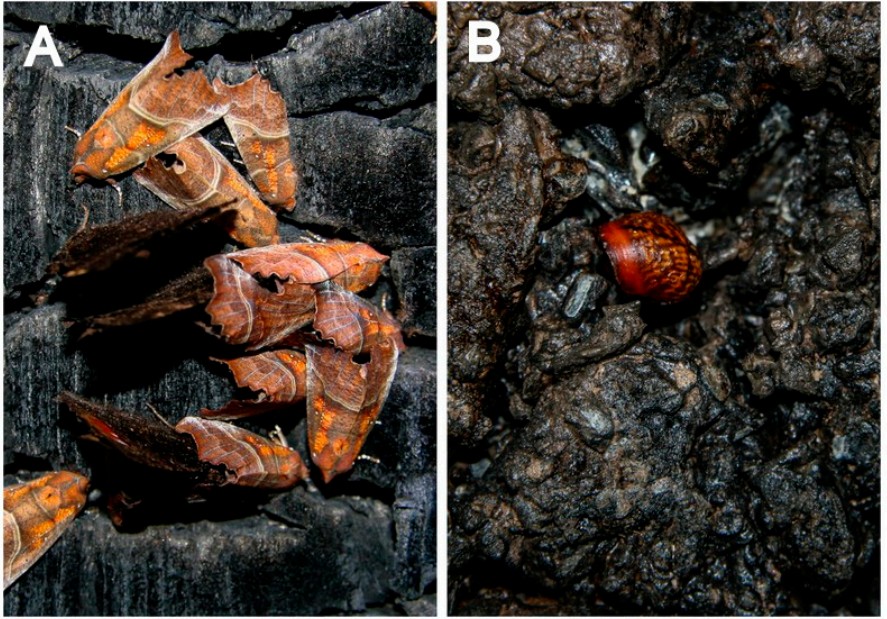

**Figure A5.** (**A**): A group of haralds (*Scoliopteryx libatrix*) on the walls of a bunker that was burned during World War II. We can observe orange drawings on the enclosed wings that resemble the letter 'M' (SA-CEF number 27). (**B**): Land snail *Monachoides incarnatus* hidden in a wall gap (SA-CEF number 37).

**Table A1.** Overview table of SA-CEFs studied objects with their description. The numbering of the individual SA-CEFs corresponds to the data in Table 1—part 1.

| No. | Name: | N: | E: | Altitude: | Military Administrative Section: | Ownership: |
|---|---|---|---|---|---|---|
| 1 | XIX/432/A-140Z | 49.7681898 | 18.5956608 | 284 | XIX Louky | personal |
| 2 | XXI/10/E | 50.1581184 | 17.3699108 | 660 | XXI Mnichov | government |
| 3 | XXI/11/B1-80Z | 50.1571897 | 17.3723533 | 647 | XXI Mnichov | not found |
| 4 | XXI/15/E | 50.1545095 | 17.3669212 | 614 | XXI Mnichov | church adm. |
| 5 | VII-3/12/B1-100 | 49.9861001 | 17.5754216 | 607 | VII-3 Hohnberg (Velký Tetřev), 3.sled | church adm. |
| 6 | VII-3/13/A-120 | 49.9907793 | 17.5681580 | 619 | VII-3 Hohnberg (Velký Tetřev), 3.sled | church adm. |
| 7 | XXI/14/A-120 | 50.1589392 | 17.3584551 | 655 | XXI Mnichov | church adm. |
| 8 | XXI/122/A-180 | 50.1593096 | 17.3555914 | 715 | XXI Mnichov | church adm. |
| 9 | XXI/123/A-140 | 50.1589694 | 17.3540345 | 747 | XXI Mnichov | church adm. |
| 10 | XI/619/E | 50.0502091 | 17.4780221 | 448 | XI Bretnov (Široká Niva) | government |
| 11 | XI/594/B1-80 | 50.0515198 | 17.4 785056 | 484 | XI Bretnov (Široká Niva) | government |
| 12 | XIX/431/A-180Z | 49.7704887 | 18.5976119 | 282 | XIX Louky | personal |
| 13 | XI/461/B2-90 | 50.0519298 | 17.4808423 | 458 | XI Bretnov (Široká Niva) | government |
| 14 | XI/463/A-140Z | 50.0554488 | 17.4758975 | 453 | XI Bretnov (Široká Niva) | government |
| 15 | X/449/A-140Z | 50.0434196 | 17.5070920 | 427 | X Nové Heřminovy | Government |
| 16 | X/450/A-160 | 50.0449989 | 17.5030057 | 416 | X Nové Heřminovy | personal |
| 17 | XVI/208/A-140 | 49.9136298 | 18.4120698 | 201 | XVI Nový Bohumín | government/ personal |
| 18 | II/16/A-120 | 49.9118497 | 17.9925046 | 247 | II Komárov | not found |
| 19 | II/10/A-120 | 49.9126396 | 17.9864304 | 245 | II Komárov | personal |
| 20 | II/5/A-140 | 49.9178202 | 17.9806877 | 237 | II Komárov | personal |
| 21 | II/4/A-160 | 49.9176098 | 17.9875375 | 236 | II Komárov | personal |
| 22 | II/7/A-120 | 49.9156297 | 17.9887584 | 236 | II Komárov | personal |

**Table A2.** Overview table of SA-CEFs studied objects with their description. The numbering of the individual SA-CEFs corresponds to the data in Table 1—part 2.

| No. | Name: | N: | E: | Altitude: | Military Administrative Section: | Ownership: |
|---|---|---|---|---|---|---|
| 23 | XIX/600/B2-80 | 49.7729196 | 18.5934953 | 278 | XIX Louky | personal |
| 24 | XVI/204/A-140 | 49.9176600 | 18.3854910 | 200 | XVI Nový Bohumín | personal |
| 25 | XVI/7/A-120 | 49.9205797 | 18.3892101 | 194 | XVI Nový Bohumín | personal |
| 26 | XVI/8/A-140Z | 49.9197700 | 18.3973602 | 196 | XVI Nový Bohumín | government |
| 27 | XVI/205/A-140 | 49.9174100 | 18.3893288 | 206 | XVI Nový Bohumín | government |
| 28 | XIX/659/B1-80 | 49.7757991 | 18.5935640 | 285 | XIX Louky | government |
| 29 | IV/911/A-160 | 49.9537 | 17.7987 | 331 | IV Milostovice | personal |
| 30 | IV/908/A-160Z | 49.9540897 | 17.8049218 | 327 | IV Milostovice | personal |
| 31 | IV/912/A-120 | 49.95193 | 17.80241 | 330 | IV Milostovice | personal |
| 32 | XIX/602/A-140 | 49.7713399 | 18.5930522 | 283 | XIX Louky | personal |
| 33 | V/1048/A-160 | 49.9619599 | 17.7262197 | 368 | V Sádek | government |
| 34 | V/958/A-140 | 49.9614186 | 17.7284309 | 373 | V Sádek | personal |
| 35 | XIX/656/E | 49.7799299 | 18.5907008 | 276 | XIX Louky | government |
| 36 | XIX/401/A-160Z | 49.8251091 | 18.5624953 | 233 | XIX Louky | not found |
| 37 | XXI/6/A-180 | 50.15867 | 17.37575 | 646 | XXI Mnichov | personal |
| 38 | XXI/2/A-140 | 50.1586885 | 17.3701837 | 661 | XXI Mnichov | government |
| 39 | XXI/1/A-120Z | 50.1583883 | 17.3715015 | 663 | XXI Mnichov | government |

**Table A3.** Table of SA-CEFs study objects with data that were obtained by field survey (2014-2021). The numbering of the individual SA-CEFs corresponds to the data in Table 1—part 1.

| No. | Azimuth of Entrance: | Exterior Environment: | Interior Environment: | Organic Material in Interior: | Anorganic Material in Interior: | Rate of Human Use: | Entrance: |
|---|---|---|---|---|---|---|---|
| 1 | 292,5/NWW | open | dry | none | none | intensively used | open |
| 2 | 135/SE | closed | dry | little | none | occasionally used | open |
| 3 | 157,5/SSE | transitional | dry | little | lot | intensively used | semi-open |
| 4 | 180/S | closed | wet | lot | lot | intensively used | open |
| 5 | 157,5/SSE | closed | flood | none | lot | occasionally used | semi-open |
| 6 | 225/SW | closed | dry | lot | none | occasionally used | open |
| 7 | 180/S | closed | wet | lot | lot | occasionally used | open |
| 8 | 157,5/SSE | closed | dry | little | none | occasionally used | semi-open |
| 9 | 157,5/SSE | transitional | dry | none | none | unused | semi-open |
| 10 | 225/SW | transitional | flood | little | little | occasionally used | semi-open |
| 11 | 202,5/SSW | transitional | dry | little | none | occasionally used | open |
| 12 | 270/W | open | flood | none | lot | unused | open |
| 13 | 270/W | transitional | dry | none | none | intensively used | open |
| 14 | 225/SW | open | dry | little | none | unused | open |
| 15 | 202,5/SSW | closed | wet | lot | little | unused | open |
| 16 | 202,5/SSW | transitional | wet | none | little | unused | semi-open |
| 17 | 202,5/SSW | open | dry | little | little | intensively used | open |

**Table A4.** Table of SA-CEFs study objects with data that were obtained by field survey (2014-2021). The numbering of the individual SA-CEFs corresponds to the data in Table 1—part 2.

| No. | Azimuth of Entrance: | Exterior Environment: | Interior Environment: | Organic Material in Interior: | Anorganic Material in Interior: | Rate of Human Use: | Entrance: |
|---|---|---|---|---|---|---|---|
| 18 | 180/S | open | dry | none | lot | intensively used | open |
| 19 | 202,5/SSW | open | wet | lot | lot | occasionally used | open |
| 20 | 180/S | open | dry | little | none | unused | semi-open |
| 21 | 180/S | open | dry | none | none | unused | open |
| 22 | 202,5/SSW | open | wet | none | lot | unused | semi-open |
| 23 | 270/W | transitional | dry | lot | little | occasionally used | open |
| 24 | 225/SW | open | dry | none | lot | occasionally used | open |
| 25 | 225/SW | open | flood | none | none | unused | open |
| 26 | 180/S | open | dry | little | little | unused | open |
| 27 | 180/S | transitional | dry | little | lot | occasionally used | open |
| 28 | 247,5/SWW | closed | dry | little | lot | occasionally used | open |
| 29 | 157,5/SSE | open | dry | none | none | unused | open |
| 30 | 180/S | open | dry | none | none | unused | open |
| 31 | 157,5/SSE | transitional | dry | none | none | unused | closed |
| 32 | 270/W | open | dry | little | none | occasionally used | open |
| 33 | 180/S | closed | dry | none | none | unused | open |
| 34 | 202,5/SSW | open | dry | none | none | unused | open |
| 35 | 247,5/SWW | closed | dry | little | little | occasionally used | semi-open |
| 36 | 225/SW | open | flood | lot | little | unused | semi-open |
| 37 | 202,5/SSW | open | wet | lot | little | intensively used | open |

**Table A5.** Table of SA-CEFs study objects with data that were obtained by field survey (2014-2021). The numbering of the individual SA-CEFs corresponds to the data in Table 1—part 3.

| No. | Azimuth of Entrance: | Exterior Environment: | Interior Environment: | Organic Material in Interior: | Anorganic Material in Interior: | Rate of Human Use: | Entrance: |
|---|---|---|---|---|---|---|---|
| 38 | 225/SW | transition | dry | little | none | occasionally used | semi-open |
| 39 | 180/S | transition | dry | little | none | occasionally used | semi-open |

**Table A6.** Species of butterflies and moths (Lepidoptera) and their abundances in each SA-CEFs in 2014, F—frequency in %, D—dominance in %.

| Species: | | *Aglais urticae* | *Agonopterix arenella* | *Agonopterix curvipunctosa* | *Agonopterix heracliana* | *Agonopterix ocellana* | *Hypena rostralis* | *Inachis io* | *Nymphalis polychloros* | *Scoliopterix libatrix* |
|---|---|---|---|---|---|---|---|---|---|---|
| | 1 | 1 | ~ | ~ | ~ | ~ | 2 | 19 | ~ | 7 |
| | 2 | ~ | ~ | ~ | ~ | ~ | ~ | 11 | ~ | 24 |
| | 3 | ~ | ~ | ~ | ~ | ~ | ~ | 24 | ~ | 12 |
| | 4 | ~ | ~ | ~ | ~ | ~ | ~ | 7 | ~ | 12 |
| | 5 | ~ | ~ | ~ | 1 | ~ | ~ | 6 | ~ | 8 |
| | 6 | ~ | ~ | ~ | ~ | ~ | ~ | ~ | ~ | 12 |
| | 7 | ~ | ~ | ~ | ~ | ~ | ~ | 2 | ~ | 11 |
| | 8 | ~ | ~ | ~ | ~ | ~ | ~ | 6 | ~ | 28 |
| | 9 | ~ | ~ | ~ | 1 | ~ | 1 | 1 | ~ | 1 |
| | 10 | ~ | ~ | ~ | ~ | ~ | 1 | 2 | ~ | 1 |
| | 11 | ~ | 1 | ~ | ~ | ~ | ~ | 8 | ~ | 7 |
| | 12 | ~ | ~ | ~ | ~ | ~ | 1 | 6 | 1 | 8 |
| | 14 | 5 | ~ | ~ | 3 | ~ | 3 | 29 | ~ | 22 |
| | 15 | ~ | ~ | ~ | ~ | ~ | ~ | 4 | ~ | 22 |
| | 16 | ~ | ~ | ~ | ~ | ~ | ~ | 7 | ~ | 8 |
| | 17 | ~ | ~ | ~ | ~ | ~ | 1 | 1 | ~ | ~ |
| | 18 | ~ | ~ | ~ | ~ | 1 | 7 | 2 | ~ | 1 |
| | 19 | 2 | ~ | ~ | ~ | ~ | ~ | 10 | ~ | 3 |
| Object number: | 20 | 1 | ~ | ~ | ~ | ~ | 1 | 1 | ~ | 1 |
| | 21 | ~ | ~ | ~ | ~ | ~ | 1 | 1 | ~ | ~ |
| | 23 | ~ | 1 | ~ | ~ | ~ | ~ | 1 | ~ | 7 |
| | 24 | ~ | ~ | ~ | 1 | ~ | 3 | 15 | ~ | 2 |
| | 25 | ~ | ~ | ~ | ~ | ~ | 5 | 8 | ~ | 7 |
| | 26 | ~ | ~ | ~ | ~ | ~ | 15 | 22 | ~ | 5 |
| | 27 | ~ | ~ | ~ | ~ | 1 | 3 | 5 | ~ | 1 |
| | 28 | ~ | ~ | ~ | ~ | ~ | ~ | 9 | ~ | 20 |
| | 29 | ~ | ~ | ~ | ~ | ~ | ~ | 6 | ~ | ~ |
| | 30 | ~ | ~ | ~ | ~ | ~ | ~ | 21 | ~ | ~ |
| | 31 | ~ | ~ | ~ | 1 | ~ | 27 | ~ | ~ | 4 |
| | 32 | 3 | ~ | ~ | ~ | ~ | 5 | 19 | 1 | 10 |
| | 33 | ~ | 3 | 1 | ~ | ~ | ~ | ~ | ~ | 2 |
| | 34 | ~ | 1 | ~ | ~ | ~ | 1 | 10 | ~ | ~ |
| | 35 | ~ | ~ | ~ | ~ | ~ | ~ | 1 | ~ | 1 |
| | 36 | ~ | ~ | ~ | ~ | ~ | ~ | 1 | ~ | 5 |
| | 37 | ~ | ~ | ~ | ~ | ~ | ~ | 7 | ~ | 18 |
| | 38 | 1 | ~ | ~ | ~ | ~ | ~ | 19 | ~ | 20 |
| | 39 | ~ | ~ | ~ | 2 | ~ | ~ | 22 | ~ | 19 |
| Σ | | 13 | 6 | 1 | 9 | 2 | 77 | 313 | 2 | 309 |
| F (%) | | 16.22 | 10.81 | 2.7 | 16.22 | 5.41 | 43.24 | 91.89 | 5.41 | 86.49 |
| D (%) | | 1.78 | 0.82 | 0.14 | 1.23 | 0.27 | 10.52 | 42.76 | 0.27 | 42.21 |

**Table A7.** Species of land snails (Gastropoda) and their abundances in each SA-CEFs in 2014, F—frequency in %, D—dominance in %.

| Species: Object number: | Aegopinella nitens | Alinda biplicata | Arianta arbustorum | Arion vulgaris | Cepaea hortensis | Cochlicopa lubrica | Discus rotundatus | Discus ruderatus | Ena montana | Fruticicola fruticum | Helix pomatia | Limax cinereoniger | Macrogastra ventricosa | Monachoides incarnatus | Monachoides vicinus | Oxychilus cellarius | Perforatella bidentata | Trochulus hispidus | Vitrea crystallina | Vitrina pellucida |
|---|---|---|---|---|---|---|---|---|---|---|---|---|---|---|---|---|---|---|---|---|
| 2 | ~ | ~ | 1 | ~ | ~ | ~ | ~ | ~ | ~ | ~ | ~ | ~ | ~ | 1 | ~ | ~ | ~ | ~ | ~ | ~ |
| 3 | ~ | ~ | 1 | ~ | ~ | ~ | ~ | ~ | ~ | ~ | ~ | ~ | ~ | 2 | ~ | ~ | ~ | ~ | ~ | ~ |
| 7 | ~ | ~ | ~ | ~ | ~ | ~ | 17 | 1 | ~ | ~ | ~ | ~ | ~ | ~ | ~ | ~ | ~ | ~ | ~ | ~ |
| 8 | ~ | ~ | ~ | ~ | ~ | ~ | ~ | 2 | ~ | ~ | ~ | ~ | ~ | ~ | ~ | ~ | ~ | ~ | 1 | ~ |
| 10 | 1 | ~ | 2 | ~ | ~ | ~ | ~ | ~ | ~ | ~ | ~ | 1 | ~ | 3 | 2 | ~ | ~ | ~ | ~ | ~ |
| 11 | ~ | ~ | 3 | ~ | ~ | ~ | ~ | ~ | ~ | ~ | ~ | ~ | ~ | 3 | ~ | ~ | ~ | ~ | ~ | ~ |
| 14 | ~ | ~ | 1 | ~ | ~ | ~ | ~ | ~ | ~ | ~ | ~ | ~ | ~ | ~ | ~ | ~ | ~ | ~ | ~ | ~ |
| 15 | ~ | ~ | 2 | ~ | 1 | ~ | ~ | ~ | 3 | ~ | ~ | ~ | ~ | 4 | ~ | ~ | 1 | 3 | ~ | ~ |
| 17 | ~ | ~ | ~ | ~ | ~ | ~ | ~ | ~ | ~ | 1 | ~ | ~ | ~ | 1 | ~ | ~ | ~ | ~ | ~ | ~ |
| 19 | ~ | ~ | ~ | ~ | 1 | ~ | ~ | ~ | ~ | 2 | 3 | ~ | ~ | ~ | ~ | ~ | ~ | ~ | ~ | ~ |
| 20 | ~ | ~ | ~ | 1 | ~ | ~ | ~ | ~ | ~ | 3 | 4 | 1 | ~ | ~ | ~ | ~ | ~ | ~ | ~ | ~ |
| 21 | ~ | ~ | ~ | ~ | ~ | ~ | ~ | ~ | ~ | ~ | 7 | ~ | ~ | 1 | ~ | ~ | ~ | ~ | ~ | ~ |
| 22 | ~ | ~ | ~ | ~ | ~ | ~ | ~ | ~ | ~ | 3 | 1 | ~ | 1 | 1 | ~ | 1 | ~ | 1 | ~ | ~ |
| 23 | ~ | 2 | ~ | ~ | ~ | ~ | ~ | ~ | ~ | ~ | ~ | ~ | ~ | 2 | ~ | ~ | ~ | ~ | ~ | 3 |
| 25 | ~ | ~ | ~ | ~ | ~ | 3 | ~ | ~ | ~ | ~ | 1 | ~ | ~ | ~ | ~ | ~ | ~ | ~ | ~ | ~ |
| 27 | ~ | ~ | ~ | ~ | ~ | ~ | ~ | ~ | ~ | ~ | 1 | ~ | ~ | 4 | ~ | ~ | ~ | ~ | ~ | ~ |
| 29 | ~ | ~ | ~ | ~ | 2 | ~ | ~ | ~ | ~ | ~ | 3 | ~ | ~ | ~ | ~ | ~ | ~ | ~ | ~ | ~ |
| 33 | ~ | 2 | ~ | ~ | ~ | ~ | ~ | ~ | ~ | ~ | 3 | ~ | ~ | 2 | ~ | ~ | ~ | ~ | ~ | ~ |
| 34 | ~ | ~ | ~ | ~ | ~ | ~ | ~ | ~ | ~ | ~ | 14 | ~ | ~ | ~ | ~ | ~ | ~ | ~ | ~ | ~ |
| 37 | 1 | 8 | 11 | ~ | ~ | ~ | 2 | ~ | ~ | ~ | 7 | ~ | ~ | 2 | ~ | 1 | ~ | 9 | 1 | ~ |
| 38 | ~ | 3 | 1 | ~ | 1 | ~ | ~ | ~ | ~ | ~ | ~ | ~ | ~ | 1 | ~ | ~ | ~ | ~ | ~ | ~ |
| 39 | ~ | ~ | ~ | ~ | ~ | ~ | ~ | ~ | ~ | ~ | ~ | ~ | ~ | 1 | ~ | ~ | ~ | ~ | ~ | ~ |
| Σ | 2 | 15 | 22 | 1 | 5 | 3 | 19 | 3 | 3 | 9 | 44 | 2 | 1 | 28 | 2 | 2 | 1 | 13 | 2 | 3 |
| F (%) | 9.09 | 18.18 | 36.36 | 4.55 | 18.18 | 4.55 | 9.09 | 9.09 | 4.55 | 18.18 | 45.45 | 9.09 | 4.55 | 63.64 | 4.55 | 9.09 | 4.55 | 13.64 | 9.09 | 4.55 |
| D (%) | 1.11 | 8.33 | 12.22 | 0.56 | 2.78 | 1.67 | 10.56 | 1.67 | 1.67 | 5 | 24.44 | 1.11 | 0.56 | 15.56 | 1.11 | 1.11 | 0.56 | 7.22 | 1.11 | 1.67 |

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
