# Peer review of "Units of Military Fortification Complex as Phenomenon Elements of the Czech Borderlands Landscape"

_land, doi:10.3390/land11010079_

Round 1

Reviewer 1 Report

This paper is focused on selected units of casemates with enhanced fortification (CEFs) in  landscape as specific forms of brownfields. The authors have carried out analyses and surveys that clearly demonstrate that the CEF performs a number of important functions in the landscape, for which they cannot be considered brownfield. Units of CEFs have a cultural and historical character, and thus create primarily historical and cultural context in the landscape. Although today it no longer fulfills the defensive function of the state, it performs a different function and does not generate any economic, ecological or social barrier or burden. CEF can even be a benefit for the landscape when used appropriately, for example education, tourism, animal hibernation, ect.

The paper is not consistent. Some chapters and subchapters are too extensive and some are insufficient. Some parts of the text do not correspond to the titles of chapters and subchapters, for example, the description of the respondents and the guided interview belong to the methods instead of the results.

  1. Introduction is clear. The terms are well-defined, objects are located in the landscape and the individual types of fortifications are well-described. I miss the reason why the authors chose the north-eastern part of the fortifications.The authors mention the uniqueness of the fortification, it would be appropriate to compare it with the fortification systems in other parts of the world.

2. Materials and Methods

line 209 -210: Due to the large total number of these objects, only a part of them was selected for the purpose of a more detailed study.  - The selection methodology needs to be complemented.

line 240: It is necessary to supplement the identification of respondents, e.g. age structure, visitors or residents, etc.

Lines 343 -349 are part of the results. I think these lines belong to the methods.

3. Results

line 369 Could you answer following questions?: How many respondents knew the stories?Is it possible to evaluate a guided interview statistically?

3.2. Selected CEFs units as brownfields

I positively evaluate the field mapping in the landscape with the additional information about ownership, using, etc.

3.3. Functional potential of selected CEFs units in the landscape

It is necessary to add information on how people perceive CEFs units in the landscape (positively or negatively and why). The same would certainly be added when evaluating CEF as a geomorphological forms (if the geomorphological forms are negative or positive in the landscape).

line 560: Table 3 - It would be appropriate to link the role of the stories to the number of respondents, how many respondents agreed with the role of stories.

4.2. Selected CEFs units as brownfields - Graph 8. The relationship between "Ownership of selected CEF instruments" and "The rate of human use in the selected CEF instruments" is very interesting and produces interesting and unexpected results. However, it would be appropriate to indicate how the CEFs are being used. If it is used in a positive way e.g. in tourism, education, or in a negative way e.g. squatter settlements. 

Author Response

We very much appreciate your comments, which we consider substantive and useful. We have tried to include all of your comments in our article without exception. However, we have significantly revised some passages based on the comments of all other reviewers. We hope that the incorporation of your comments can still be traced in the revised article.

We tried to reduce the length of the whole text and connect individual parts too.

Your whole review was a great encouragement, a pleasure and a motivation for our further research in this area.

response:

This paper is focused on selected units of casemates with enhanced fortification (CEFs) in landscape as specific forms of brownfields. The authors have carried out analyses and surveys that clearly demonstrate that the CEF performs a number of important functions in the landscape, for which they cannot be considered brownfield. Units of CEFs have a cultural and historical character, and thus create primarily historical and cultural context in the landscape. Although today it no longer fulfills the defensive function of the state, it performs a different function and does not generate any economic, ecological or social barrier or burden. CEF can even be a benefit for the landscape when used appropriately, for example education, tourism, animal hibernation, ect.

The paper is not consistent. Some chapters and subchapters are too extensive and some are insufficient. Some parts of the text do not correspond to the titles of chapters and subchapters, for example, the description of the respondents and the guided interview belong to the methods instead of the results.

There has been an overall reduction of the article from 42 to 33. The positioning of some parts of the text has been reassessed to match the chapter and subchapter titles.

  1. Introduction is clear. The terms are well-defined, objects are located in the landscape and the individual types of fortifications are well-described. I miss the reason why the authors chose the north-eastern part of the fortifications. The reason for choosing the north-eastern part of the fortification is given in chapter 2.2. Socio-economic sphere, from line 257.

The authors mention the uniqueness of the fortification, it would be appropriate to compare it with the fortification systems in other parts of the world. Examples of fortifications from around the world are given in the introduction.

  1. Materials and Methods

line 209 -210: Due to the large total number of these objects, only a part of them was selected for the purpose of a more detailed study.  - The selection methodology needs to be complemented.

The selection methodology is corrected - page 7, lines 270-291.

line 240: It is necessary to supplement the identification of respondents, e.g. age structure, visitors or residents, etc. Added identification of respondents - page 6 (2.1. Genius loci), lines 234-236.

Lines 343 -349 are part of the results. I think these lines belong to the methods. Lines have been moved to methods - page 6 (2.1. Genius loci), lines 229-236.

  1. Results

line 369 Could you answer following questions?: How many respondents knew the stories? All respondents ultimately understood what we mean by 'stories' and the meaning we attach to them. It is true, however, that for some this was only after our follow-up questions (which is why we chose the guided interview method). Then they were all able to state the stories, distinguishing them from historical events. Most, however, did not fully understand the contribution of the stories, considering them as useless ballast. It was only when we outlined that 'these stories' also had their importance that they were willing to admit their contribution, but with the caveat: 'it should be done in moderation'; 'some stories do us a disservice', etc. We mention this in the discussion.

However, we now realize that we should have made this explicit in the article, for which we apologize. Due to the length of the post and the number of reviews, we forgot to include this important fact in the article.

lines 227-231: ‘In order to capture this potential, a guided interview method was chosen, with respondents were asked about the stories associated with A-CEFs (including capturing the wider context associated with the place). Due to the specific topic of the research, it was necessary to approach suitable respondents who are in some way affected by the genius loci and the associated function of A-CEFs in the landscape.’

lines 529-534: ‘The guided interview as a research method was not focused on the general public at this stage, but mainly at people who were expected to have some knowledge of the issue - people who dealing with this issue in their profession (museum workers) and people who are involved in this field in their free time. For the guided interview, which is time consuming, the number of respondents (27) was not very high, but for our research it is sufficient.’

Is it possible to evaluate a guided interview statistically? From these results, the guided interview cannot be statistically evaluated. Explained on page 15 (4.1. Genius loci), lines 542-546.

3.2. Selected CEFs units as brownfields

I positively evaluate the field mapping in the landscape with the additional information about ownership, using, etc.

3.3. Functional potential of selected CEFs units in the landscape

It is necessary to add information on how people perceive CEFs units in the landscape (positively or negatively and why). The same would certainly be added when evaluating CEF as a geomorphological forms (if the geomorphological forms are negative or positive in the landscape). Mentioned in chapter 4.1. Genius loci - page 16, lines 580-584.

line 560: Table 3 - It would be appropriate to link the role of the stories to the number of respondents, how many respondents agreed with the role of stories. As mentioned above in the comments on this issue.

4.2. Selected CEFs units as brownfields - Graph 8. The relationship between "Ownership of selected CEF instruments" and "The rate of human use in the selected CEF instruments" is very interesting and produces interesting and unexpected results. However, it would be appropriate to indicate how the CEFs are being used. If it is used in a positive way e.g. in tourism, education, or in a negative way e.g. squatter settlements.

lines 575-578: At this point, it is also worth highlighting that, apparently due to the occurrence of SA-CEFs in the open landscape (outside human settlements), we have not observed negative uses (e.g. squatter settlements).

lines 758-761: The common and typical feature of these A-CEFs is that they are all part of the cultural heritage (although they are not listed as heritage sites, e.g., by UNESCO), but for obvious reasons, there is not enough interest in their use (a large number of them). 

Reviewer 2 Report

In general, the paper is interesting. The empirical parts contain good information on may aspects of the fortifications, from insects to stories. 

I have some problems with the Introduction, that seems based on a substantial amount of research that is, however, unknown outside Czechia. What exactly is the research question? What is the reason for writing this article? Is it interest in landscape, or in history, or in heritage, or in participation? This could be stated much clearer.

Line 31: number one topic: where?; in which?

Lines 38-40: “As a result of anthropogenic influence on the original natural landscape, a new char-38 acter, a new spirit of the landscape, and a new way of looking at the landscape is being 39 created [7].” But what exactly is meant here? What spirit? What way of looking?

Line 42: Four structures. This is in fact a traditional view, elsewhere also described as ‘layers’, in which the natural landscape is the starting point and human activities take place within natural frameworks. Of parts of the information, for example the section on climate (Line 219 etc.) the relevance is not very clear to me.

Lines 58-59: “In general, the meaning of [the] term 58 'brownfield' is an object (area or building) that has lost its original functional potential.” I would limit the term to former urban, mining or industrial uses (when agricultural sites are named as brownfields, as in line 62, the term loses its meaning). Also the term “original” is problematic (what is original?). Furthermore, in urban planning (see Wikipedia ‘brownfield land’) brownfields are defined as: any previously developed land that is not currently in use that may be potentially contaminated. So it seems good to rethink the definition of brownfield.

Line 80: What exactly do you mean with the term ‘casemates with enhanced fortification’? Do you suggest that there are also casemates without military functions?

Line 114: Please show this region on the map.

Line 238: ‘The genius loci of the place’ is double.

There are two final remarks:

1 I miss an international perspective. The reuse and management of derelict military fortifications is an international theme of study. The authors could, for example, look at military fortifications on the Word Heritage List (Vauban-fortifications in France, the fortresses around Amsterdam, European fortifications in Ghana etc.) or the Atlantikwall that is now also partly protected. Many of these protected historic fortifications have a recent history of creative re-use.

2 The theme of heritage is not defined or elaborated. Adapted reuse is a central theme within heritage studies. See (to mention just one example) DeSilvey, Curated decay. In this respect, it would also be important to connect the present meaning of the fortifications to their origin as a self-conscious fortification to defend the young country and one of the last surviving democracies in Central Europe against upcoming aggressors.

Author Response

We very much appreciate your comments, which we consider substantive and useful. We have tried to include all of your comments in our article without exception. However, we have significantly revised some passages based on the comments of all other reviewers. We hope that the incorporation of your comments can still be traced in the revised article.

We tried to better define the issue of brownfields and also to frame the problematic of fortification units in an international context.

Your whole review was a great encouragement, a pleasure and a motivation for our further research in this area.

response:

In general, the paper is interesting. The empirical parts contain good information on may aspects of the fortifications, from insects to stories.

I have some problems with the Introduction, that seems based on a substantial amount of research that is, however, unknown outside Czechia. What exactly is the research question? What is the reason for writing this article? Is it interest in landscape, or in history, or in heritage, or in participation? This could be stated much clearer. Already in the introduction, we tried to clarify the subject (objectives) of our paper. The main objective is the issue of brownfields as one of the main environmental issues.

We believe that the issue of brownfields should be seen from a different perspective. It does not seem to us to be the most appropriate when the definitions of brownfields underline their insignificance... In our example, we want to show that brownfields can actually fulfill a number of positive functions as brownfields.

Line 31: number one topic: where?; in which? This discrepancy has been removed from the text. The Introduction has been completely rewritten - also because of the above and the comments of other reviewers.

Lines 38-40: “As a result of anthropogenic influence on the original natural landscape, a new char-38 acter, a new spirit of the landscape, and a new way of looking at the landscape is being 39 created [7].” But what exactly is meant here? What spirit? What way of looking? This is probably a mistake due to translation. We have now tried to be more consistent with the term "genius loci" throughout the article. The Czech language is more "colorful" in its expressions and unfortunately could be sometimes unclear as a result.

Line 42: Four structures. This is in fact a traditional view, elsewhere also described as ‘layers’, in which the natural landscape is the starting point and human activities take place within natural frameworks. Of parts of the information, for example the section on climate (Line 219 etc.) the relevance is not very clear to me. Reworked, ‘structures’ were deleted and replaced with ‘layers’. Many parts were shortened, including the climate passage, it was really unnecessarily lengthy.

Lines 58-59: “In general, the meaning of [the] term 58 'brownfield' is an object (area or building) that has lost its original functional potential.” I would limit the term to former urban, mining or industrial uses (when agricultural sites are named as brownfields, as in line 62, the term loses its meaning). Also the term “original” is problematic (what is original?). Furthermore, in urban planning (see Wikipedia ‘brownfield land’) brownfields are defined as: any previously developed land that is not currently in use that may be potentially contaminated. So it seems good to rethink the definition of brownfield. Deleted, revised. We have now tried to work more carefully with the term 'brownfields', including its definition.

Line 80: What exactly do you mean with the term ‘casemates with enhanced fortification’? Do you suggest that there are also casemates without military functions? Revised, the abbreviations CEFs, A-CEFs and SA-CEFs have been incorporated into the text, which now allows better orientation in the issue of when we are dealing with objects with an old function (military, CEFs) and when we are dealing with abandoned, brownfields objects (abandoned, A-CEFs), or objects that we specifically focused on during our research (SA-CEFs).

Line 114: Please show this region on the map. Has been incorporated into the article.

Line 238: ‘The genius loci of the place’ is double. It has been redesigned.

There are two final remarks:

1 I miss an international perspective. The reuse and management of derelict military fortifications is an international theme of study. The authors could, for example, look at military fortifications on the Word Heritage List (Vauban-fortifications in France, the fortresses around Amsterdam, European fortifications in Ghana etc.) or the Atlantikwall that is now also partly protected. Many of these protected historic fortifications have a recent history of creative re-use.

We have now tried to frame the issue in a broader framework with newly incorporated examples. Adaptive reuse strategies are not our aim of the research. A-CEF loses its hidden singularity in case of its re-use (lines: 781-791).

2 The theme of heritage is not defined or elaborated. Adapted reuse is a central theme within heritage studies. See (to mention just one example) DeSilvey, Curated decay. In this respect, it would also be important to connect the present meaning of the fortifications to their origin as a self-conscious fortification to defend the young country and one of the last surviving democracies in Central Europe against upcoming aggressors.

By refining the objectives and reworking the introduction, we have attempted to place 'cultural heritage' in the context in which we approach it in our work. Where A-CEF plays only a peripheral role.

Lines: 66-77

The places where we introduced the term 'cultural heritage' in the previous version of the paper were due to the ambiguity of the term, which is perceived more broadly and generally in the Czech language.

Reviewer 3 Report

The paper is methodologically well set up and includes several methods - such as mapping observation, interviews and analysis. The biggest drawback is the length of the article itself. Although there are numerous contributions, some of them should be omitted, and repetitive parts of the text are unnecessary (e.g. some claims are repeated in both the results and the discussion). Some photos are redundant too.

Also, it is not clear why are the structures given in text, explained from quaternary to primary – which is confusing for the reader and makes it difficult to follow the thoughts of the authors.

At the begging of paper authors write about planned casemates (“In the Moravian-Silesian region alone (in terms 205 of the current administrative structure of the Czech Republic), approximately 1,000 CEFs were planned“) and than later bring exact naumbers. This is necessary. Authors should give exact numbers of casamates (in Czech R. And Moravian-Silesian region at the begginig). This way it wouldn't be confusing.

Also, when reading about second approach it isn’t quite clear why certain casemates (39 of them) are taken into analysis, and not some others. This should be better explained.

It should be avoided to write …see graph 3 or see graph 5.

In this context, the article can be considered an original scientific paper. References, although most of them older than last 5 years (only 7 or 8 of them are published after 2015), are correctly given except for some -  QGIS software is not a source and as such cannot be cited (as it is primarily a tool for cartographic representation). The same thing is with quoting both Microsoft Excel and R.Core Team software - these are not sources or literature. Other than this I didn’t notice self-citations and other unappropriated citations.

Paper is written in scientific manner and in appropriate way. The results qualify for high standards of presentation since they are obtained from period 2014 until 2021 and they are presented in several different methods (used in human and natural disciplines). The one thing that should be improved when taking the quality of presentation in consideration, are the maps from Fig. A2-A10. Although the lines with casemates are generally shown in map above (Fig. A1), for the readers of the paper who are not familiar with the geography of Moravian-Silesian region, these maps should be broadened with general map of this region in the right corner.  That smaller map should than show, the exact part analysed in the main map.

English terminology can be improved in several places (for example what hidden curriculum of landscape means?), but in general it is acceptable.

Author Response

We very much appreciate your comments, which we consider substantive and useful. We have tried to include all of your comments in our article without exception. However, we have significantly revised some passages based on the comments of all other reviewers. We hope that the incorporation of your comments can still be traced in the revised article.

We have tried to make the whole text shorter and clearer. We also worked on removing confusing parts in the text.

Your whole review was a great encouragement, a pleasure and a motivation for our further research in this area.

response:

The paper is methodologically well set up and includes several methods - such as mapping observation, interviews and analysis. The biggest drawback is the length of the article itself. Although there are numerous contributions, some of them should be omitted, and repetitive parts of the text are unnecessary (e.g. some claims are repeated in both the results and the discussion). Some photos are redundant too.

Some parts of the article were removed (pictures and repetitive text also). The article was shortened from 42 pages to 33.

Also, it is not clear why are the structures given in text, explained from quaternary to primary – which is confusing for the reader and makes it difficult to follow the thoughts of the authors.

The term “structure” was edited to “layers” and parts of the text have been shortened by the definitions of structures. The text is clearer now.

At the begging of paper authors write about planned casemates (“In the Moravian-Silesian region alone (in terms 205 of the current administrative structure of the Czech Republic), approximately 1,000 CEFs were planned“) and than later bring exact naumbers. This is necessary. Authors should give exact numbers of casamates (in Czech R. And Moravian-Silesian region at the begginig). This way it wouldn't be confusing.

We give exact numbers of CEFs in Moravian-Silesian Region into the Materials and Methods. Whole description of fortification in the Czech Republic and especially in the Moravian-Silesian Region is on lines 196-199.

Also, when reading about second approach it isn’t quite clear why certain casemates (39 of them) are taken into analysis, and not some others. This should be better explained.

This is now better explained in Materials and Methods, lines 270-307.

It should be avoided to write …see graph 3 or see graph 5.

Edited in the whole article.

In this context, the article can be considered an original scientific paper. References, although most of them older than last 5 years (only 7 or 8 of them are published after 2015), are correctly given except for some -  QGIS software is not a source and as such cannot be cited (as it is primarily a tool for cartographic representation). The same thing is with quoting both Microsoft Excel and R.Core Team software - these are not sources or literature. Other than this I didn’t notice self-citations and other unappropriated citations.

We reduced all program resources (Excell, QGis, Rcore) and some publications older than 2015. On the other hand, we changed a few resources (namely numbers 19,20,40) and added a few new resources. In the new article, these are numbers - 1, 4, 5, 6, 57, 58, 59.

In the case of the ‘hidden curriculum’ of the landscape or brownfields, we have not avoided one self-citation, but this is only because we wanted to prove the use of this term. If this would be unsuitable, and only the definition of 'hidden curriculum' as it is now given in the paper would suffice, we will remove it.

Paper is written in scientific manner and in appropriate way. The results qualify for high standards of presentation since they are obtained from period 2014 until 2021 and they are presented in several different methods (used in human and natural disciplines). The one thing that should be improved when taking the quality of presentation in consideration, are the maps from Fig. A2-A10. Although the lines with casemates are generally shown in map above (Fig. A1), for the readers of the paper who are not familiar with the geography of Moravian-Silesian region, these maps should be broadened with general map of this region in the right corner.  That smaller map should than show, the exact part analysed in the main map.

Maps with selected sections from (original) Figure A2-A10 were added into the collage (now Figure A3. The title of this figure contains a link to Figure A2 where all selected sections in Moravian-Silesian Region are shown.

English terminology can be improved in several places (for example what hidden curriculum of landscape means?), but in general it is acceptable.

Terms in the article were clearly defined:

  • ‘hidden curriculum of landscape’: line 149-157
  • ‘landscape singularity’ and ‘hidden singularity’: lines 751-782
  • ‘structure’ replaced by ‘layers’, removing the terms primary to secondary which confused the potential reader

Reviewer 4 Report

The paper has implemented a very detailed research on the post-military landscape. This remands me the Great Wall in China which has also lost its defense function, but it is definitely not a brownfield. The authors have explored the fortifications regarding to the stories in the background, their current utilization condition, functional potential, and their inhabitant. Very detailed field survey has done for collecting these data and the results can be used for understanding the typical landscape on the border area in Czech Republic. As a semi-natural ecosystem the fortification can supply ecosystem services (e.g. recreation, education, etc.) to human and support biodiversity. However, there are still some issues need to be further clarified. In general, I would recommend major revisions.

  1. Introduction: the main shortage is the lack of the research objective in this section. Although some partial objectives are mentioned in Material and Methods section, a clear statement of the research objective (or a set of objectives) needed to presented in the end of Introduction. Therefore, the audiences can better follow the whole article.
  2. Materials and Methods: the structure could be changed. Four approaches are introduced at the beginning, but this part is somehow redundant. A brief summary about the approaches can be introduced in the introduction part with corresponding objective. Then, in the methods part the first would be the research site introduction. There are also some redundant parts in the text, e.g. sentences from line 286-289 are repetitive, Figure 4 could be combined with Figure 3. I also don’t understand how the authors choose the 39 CEFs. For section A to I how many CEFs are needed?
  3. Results: the authors have mentioned the the type: CEFs, heavy fortifications, artillery forts. But I cannot understand the type column in Appendix B. It seems the type attribution is not applied in the analysis. Maybe you can delete it. Regarding to the functional potential, the authors only give some examples about the selected CEFs, not a result for all CEFs. This could be important to zoological survey. I cannot also see the aim of classification of fortifications. Further explanation should be added.
  4. Discussion: “open” on the horizontal axis of Graph 9 should be “transitional”. Vertical axis and horizontal axis of Graph 10 should be exchanged. In this part, authors only discussed the results of each approach, but a joint analysis of the results from different approach could be more meaningful. Therefore, I recommend a joint analysis of the results from different approach, especially from approach 3 and 4. 
  5. In general, the authors have done a very great documentary work about the fortifications. With deeper data mining, more contributions could be drawn for the protection and restoration of the post-military landscape, for example, different conservation strategies should be adopted based on the classification of CEFs.

Author Response

We very much appreciate your comments, which we consider substantive and useful. We have tried to include all of your comments in our article without exception. However, we have significantly revised some passages based on the comments of all other reviewers. We hope that the incorporation of your comments can still be traced in the revised article.

We have tried to make the whole text shorter and clearer. We also worked on removing confusing parts in the text. We also tried to better define the subject of our study in the Introduction.

Your whole review was a great encouragement, a pleasure and a motivation for our further research in this area.

response:

The paper has implemented a very detailed research on the post-military landscape. This remands me the Great Wall in China which has also lost its defense function, but it is definitely not a brownfield. The authors have explored the fortifications regarding to the stories in the background, their current utilization condition, functional potential, and their inhabitant. Very detailed field survey has done for collecting these data and the results can be used for understanding the typical landscape on the border area in Czech Republic. As a semi-natural ecosystem the fortification can supply ecosystem services (e.g. recreation, education, etc.) to human and support biodiversity. However, there are still some issues need to be further clarified. In general, I would recommend major revisions.

  1. Introduction: the main shortage is the lack of the research objective in this section. Although some partial objectives are mentioned in the Material and Methods section, a clear statement of the research objective (or a set of objectives) needed to present in the end of Introduction. Therefore, the audience can better follow the whole article. The research objective was added to the introduction - page 4, line 140. The sub-objectives are listed in the individual approaches in the introduction - pages 4-5, lines 145-183.

  1. Materials and Methods: the structure could be changed. Four approaches are introduced at the beginning, but this part is somehow redundant. A brief summary about the approaches can be introduced in the introduction part with corresponding objective. Then, in the methods part the first would be the research site introduction. There are also some redundant parts in the text, e.g. sentences from line 286-289 are repetitive, Figure 4 could be combined with Figure 3. I also don’t understand how the authors choose the 39 CEFs. For section A to I how many CEFs are needed?

Four approaches are introduced in Introduction now. All repetitive parts were reduced or deleted. All redundant parts were reduced.

Selection of 39 objects was better explained in Materials and Methods, lines 270-307.

Figure 3 from the original manuscript was deleted.

  1. Results: the authors have mentioned the the type: CEFs, heavy fortifications, artillery forts. But I cannot understand the type column in Appendix B. It seems the type attribution is not applied in the analysis. Maybe you can delete it. Regarding to the functional potential, the authors only give some examples about the selected CEFs, not a result for all CEFs. This could be important to zoological survey. I cannot also see the aim of classification of fortifications. Further explanation should be added.

Classifications of ‘type’ and ‘CEF No.’ in Appendix B were deleted.

We have incorporated abbreviations into the text, which hopefully makes the text more understandable. In the case of CEFs, these are objects with an original military function. In the case of A-CEFs, they are objects that have been abandoned or objects that have been specifically selected to our research (SA-CEFs). We think that now it will also be more obvious when it is possible to generalize the knowledge gained from the study of SA-CEFs to all A-CEFs.

  1. Discussion: “open” on the horizontal axis of Graph 9 should be “transitional”. Vertical axis and horizontal axis of Graph 10 should be exchanged. In this part, authors only discussed the results of each approach, but a joint analysis of the results from different approach could be more meaningful. Therefore, I recommend a joint analysis of the results from different approach, especially from approach 3 and 4. In the end, we decided to swap the axes - thank you for the comment - and incorporate the graphs into one image. We think it will be clearer this way.

  1. In general, the authors have done a very great documentary work about the fortifications. With deeper data mining, more contributions could be drawn for the protection and restoration of the post-military landscape, for example, different conservation strategies should be adopted based on the classification of CEFs.

We have tried to clarify the issue we are dealing with in this paper with the more precise objectives stated in the introduction and the classification of strength objects (CEF, A-CEF, SA-CEF). Restoration or adaptive reuse strategies are not our aims of the research. A-CEF loses its hidden singularity in case of its re-use (lines: 781-791).

In the future, we would like to deal with the issue of conservation but from the point of view of nature and landscape conservation (lines 764-766).

Round 2

Reviewer 2 Report

The authors have intensively revised the paper, that I feel is now ready for publication

Author Response

Thank you very much.

Your approach is an encouragement to us to do further research in this area.

We made some English changes, and we hope that the text is clearer now.

Reviewer 4 Report

All my questions have been addressed in the revised manuscript. I think the language and format can be improved, maybe by a native speaker. Several comments can be found in the attached PDF file. 

Best regards,

Author Response

All my questions have been addressed in the revised manuscript. I think the language and format can be improved, maybe by a native speaker. Several comments can be found in the attached PDF file. Thank you very much.

response:

All of your comments were accessed and integrated into the text. Thank you for pointing out any omissions in the text that we have overlooked. We made some English changes by native speaker, and we hope that the text is clearer now.

Your approach is an encouragement to us to do further research in this area.